# Plectin-mediated cytoskeletal crosstalk as a target for inhibition of hepatocellular carcinoma growth and metastasis

Zuzana Outla[1], Gizem Oyman-Eyrilmez[1], Katerina Korelova[1], Magdalena Prechova[1], Lukas Frick[2], Lenka Sarnova[1], Piyush Bisht[1], Petra Novotna[1], Jan Kosla[1,3], Patricia Bortel[4], Yasmin Borutzki[5], Andrea Bileck[4,6], Christopher Gerner[4,6], Mohammad Rahbari[3,7], Nuh Rahbari[8], Emrullah Birgin[8], Bibiana Kvasnicova[9], Andrea Galisova[10], Katerina Sulkova[10], Andreas Bauer[11], Njainday Jobe[12], Ondrej Tolde[12], Eva Sticova[13,14], Daniel Rösel[12], Tracy O'Connor[15], Martin Otahal[9], Daniel Jirak[10], Mathias Heikenwälder[3], Gerhard Wiche[16], Samuel M Meier-Menches[4,5,6], Martin Gregor[1]*

[1]Laboratory of Integrative Biology, Institute of Molecular Genetics of the Czech Academy of Sciences, Prague, Czech Republic; [2]Institute of Molecular Cancer Research, University of Zurich, Zurich, Switzerland; [3]Division of Chronic Inflammation and Cancer, German Cancer Research Center, Im Neuenheimer Feld, Heidelberg, Germany; [4]Department of Analytical Chemistry, University of Vienna, Vienna, Austria; [5]Institute of Inorganic Chemistry, University of Vienna, Vienna, Austria; [6]Joint Metabolome Facility, Medical University of Vienna and University of Vienna, Heidelberg, Germany; [7]Department of Surgery, University Hospital Mannheim, Medical Faculty Mannheim, University of Heidelberg, Mannheim, Germany; [8]Department of General and Visceral Surgery, Ulm University Hospital, Ulm, Germany; [9]Department of Natural Sciences, Faculty of Biomedical Engineering, Czech Technical University in Prague, Prague, Czech Republic; [10]Department of Radiodiagnostic and Interventional Radiology, Institute for Clinical and Experimental Medicine, Prague, Czech Republic; [11]Department of Physics, University of Erlangen-Nuremberg, Erlangen, Germany; [12]Department of Cell Biology, Faculty of Science, Charles University, BIOCEV, Prumyslova, Vestec, Czech Republic; [13]Department of Clinical and Transplant Pathology, Institute for Clinical and Experimental Medicine, Prague, Czech Republic; [14]Department of Pathology, Third Faculty of Medicine, Charles University, Prague, Czech Republic; [15]Department of Biology, North Park University, Chicago, United States; [16]Department of Biochemistry and Cell Biology, Max F. Perutz Laboratories, University of Vienna, Vienna, Austria

*For correspondence: martin.gregor@img.cas.cz

Competing interest: The authors declare that no competing interests exist.

## eLife Assessment

This **valuable** study investigated the role of PLECTIN, a cytoskeletal crosslinker protein, in hepatocellular carcinoma development and progression. Using a liver-specific Plectin knockout mouse model, the authors showed **solid** evidence that PLECTIN is critical for hepatocarcinogenesis, since inhibition of PLECTIN suppressed tumor formation in multiple models. They also show that PLECTIN is key for HCC invasion and metastasis. They show a correlation between PLECTIN inhibition and attenuated FAK, MAPK/ERK, and PI3K/AKT signaling.

**Abstract** The most common primary malignancy of the liver, hepatocellular carcinoma (HCC), is a heterogeneous tumor entity with high metastatic potential and complex pathophysiology. Increasing evidence suggests that tissue mechanics plays a critical role in tumor onset and progression. Here, we show that plectin, a major cytoskeletal crosslinker protein, plays a crucial role in mechanical homeostasis and mechanosensitive oncogenic signaling that drives hepatocarcinogenesis. Our expression analyses revealed elevated plectin levels in liver tumors, which correlated with poor prognosis for HCC patients. Using autochthonous and orthotopic mouse models we demonstrated that genetic and pharmacological inactivation of plectin potently suppressed the initiation and growth of HCC. Moreover, plectin targeting potently inhibited the invasion potential of human HCC cells and reduced their metastatic outgrowth in the lung. Proteomic and phosphoproteomic profiling linked plectin-dependent disruption of cytoskeletal networks to attenuation of oncogenic FAK, MAPK/Erk, and PI3K/Akt signatures. Importantly, by combining cell line-based and murine HCC models, we show that plectin inhibitor plecstatin-1 (PST) is well-tolerated and potently inhibits HCC progression. In conclusion, our study demonstrates that plectin-controlled cytoarchitecture is a key determinant of HCC development and suggests that pharmacologically induced disruption of mechanical homeostasis may represent a new therapeutic strategy for HCC treatment.

## Introduction

Mounting evidence indicates that tissue mechanics plays a pivotal role in cancer cell and stromal cell behavior. Tumor progression is typically associated with a pathological increase of tissue stiffness caused by excessive deposition, crosslinking, and aberrant organization of dense extracellular matrix (ECM) fibers. Increasing tissue rigidity drives tumor invasion and malignancy and correlates with poor patient survival (*Broders-Bondon et al., 2018*; *Piersma et al., 2020*).

At the cellular level, both tumor and stromal cells respond to altered mechanical properties of the extracellular milieu by translating physical cues into mechanosensitive signaling pathways. This conversion relies on focal adhesions (FAs), clusters of integrin receptors facilitating the link between the ECM and the cytoskeleton. Integrin-mediated adhesion induces the activation of FAK, MAPK/Erk, and PI3K/Akt pathways, leading to increased cell survival, migration, and invasion (*Cooper and Giancotti, 2019*; *Hoxhaj and Manning, 2020*; *Sun et al., 2016*). Subsequent activation of Rho-dependent pathways results in higher cytoskeletal tension and force transmission across FAs, thus establishing a mechanical reciprocity between ECM viscoelasticity and actomyosin-generated cytoskeletal tension. Importantly, many genes encoding components of the ECM-cytoskeletal axis and their regulators (e.g. *ACTA2*, *ITGB1*, *LMNA*, *ROCK*, and *COL* genes) are controlled by tension-dependent transcription (*Dupont et al., 2011*; *Esnault et al., 2014*). This creates a difficult-to-break positive feedback loop leading to cellular and matrix stiffening, further promoting the aggressive, pro-proliferative, and invasive tumor cell phenotype.

Emerging therapeutic strategies aimed at tumor mechanics and mechanotransduction include the targeting of the ECM and ECM modulators (e.g. lysyl oxidase and angiotensin), depletion of stromal myofibroblasts, and integrin receptors (*Piersma et al., 2020*; *Cooper and Giancotti, 2019*). Other approaches target cytoskeleton-mediated downstream cellular response to tissue stiffening (e.g. Rho-dependent actomyosin-generated contractile forces [*Bustelo, 2018*]). We hypothesized that another efficacious strategy could be the inactivation of cytoskeletal crosslinker proteins (so-called cytolinkers) (*Bouameur et al., 2014*; *Prechova et al., 2023*), large proteins of the plakin protein family, responsible for maintaining the cellular architecture. The best-studied example, a prototypical cytolinker plectin is a well-established regulator of cellular tensional homeostasis and mechanotransduction (*Prechova et al., 2023*) which is upregulated in various tumors (*Gundesli et al., 2023*; *Perez et al., 2021a*). Through its canonical actin-binding domain (ABD; *Andrä et al., 1998*) and intermediate filament (IF)-binding domain (IFBD; *Nikolic et al., 1996*), plectin crosslinks actin with IF networks and recruits them to cell adhesions, including FAs. Plectin deletion or mutation results in cytoskeletal reconfiguration accompanied by altered mechanical properties, such as cellular stiffness, stress propagation, and traction force generation (*Eisenberg et al., 2013*; *Na et al., 2009*; *Osmanagic-Myers et al., 2015*; *Prechova et al., 2022*). In addition, plectin-dependent changes in cell adhesions (*Prechova et al., 2022*; *De Pascalis et al., 2018*; *Gregor et al., 2014*; *Wang et al., 2020*) and cytoskeletal tension (*Osmanagic-Myers et al., 2015*; *Prechova et al., 2022*; *Gregor et al., 2014*; *Wang et al.,*

*2020*; *Marks et al., 2022*) are associated with aberrant integrin-mediated mechanosignaling (*Gregor et al., 2014*; *Wang et al., 2020*). Although multiple reports have linked plectin with tumor malignancy (*Perez et al., 2021a*) and other pathologies (*Prechova et al., 2023*; *Vahidnezhad et al., 2022*), mechanistic insights into how plectin functionally contributes to carcinogenesis remain largely unknown.

A malignancy with a well-known link to the overproduction of ECM components is hepatocellular carcinoma (HCC), the most common type of liver cancer. Repeated rounds of hepatocyte damage and renewal due to a number of etiologies, most commonly chronic viral infection, alcohol abuse, or a diet rich in fats and sugars, create a pro-inflammatory environment in the liver. Activated hepatic stellate cells adopt a myofibroblast phenotype and increase the production and deposition of ECM components leading to liver fibrosis which can eventually progress to liver cirrhosis. Up to 90% of HCC cases occur on a background of liver fibrosis or cirrhosis, suggesting a causal link between increased deposition of ECM components and liver carcinogenesis. Consistent with this idea, plectin mRNA has been found to be upregulated in liver carcinomas, especially in the later stages of disease (*Gundesli et al., 2023*). Thus, changes in the interactions between the cytoskeleton and ECM may be important in HCC progression, particularly during the transition from local to metastatic malignancy.

Here, we explore the role of plectin in the development and dissemination of HCC. Using publicly available HCC sequencing data and biopsies from HCC patients we identify plectin as a novel HCC marker associated with a malignant phenotype and poor survival. To explore the role of plectin in hepatocarcinogenesis, we use a genetic mouse model with liver-specific plectin ablation (*PlecΔAlb*). In this model, plectin deficiency suppresses tumor initiation and growth. We further demonstrate that CRISPR/Cas9-engineered human HCC cell lines with inactivated plectin display limited migration, invasion, and anchorage-independent proliferation which correlates with their reduced metastatic outgrowth in the lung. By comprehensive proteomic analysis, we show that plectin inactivation attenuates oncogenic FAK, MAPK/Erk, and PI3K/Akt signaling signatures. Finally, our work identifies the ruthenium-based plecstatin-1 (PST), as a candidate drug that can mimic the genetic ablation of plectin, thus providing a robust preclinical proof-of-concept for PST in the treatment of HCC. Our study implicates plectin as a potent driver of HCC, highlights its importance in metastatic spread, and points to potential novel treatment options.

## Results

### Plectin levels are elevated in HCC and predict a poor prognosis

Using 17 distinct HCC patient datasets, we confirmed that *plectin* gene (*PLEC*) expression is consistently and significantly increased in HCC samples when compared to non-tumor (NT) liver tissues (*Figure 1A*). The analysis of data from The Cancer Genome Atlas (TCGA) confirmed elevated *PLEC* expression in HCC, irrespective of HCC etiology or gender (*Figure 1—figure supplement 1A–D*). To assess whether high plectin expression is typical for a specific subpopulation of HCC patients, we created t-SNE plots and compared plectin expression patterns with those of molecular subclasses of Dr. Chiang's and Dr. Boyault's classification (*Chiang et al., 2008*; *Boyault et al., 2007*). Although we observed local clusters of patients with higher or lower *PLEC* expression levels, they did not seem to be associated with any of the largest clusters or subgroups (*Figure 1—figure supplement 1E*). Strikingly, using higher tertile expression as the cut-off, higher *PLEC* mRNA levels were associated with a significant decrease in recurrence-free survival (*Figure 1B*). A similar trend was observed across eight distinct HCC datasets (*Figure 1—figure supplement 1F*).

Consistent with expression analysis, quantitative immunofluorescence microscopy of 19 human HCC tissue sections revealed a significant increase of plectin fluorescence intensities in tumor (T) compared to adjacent non-tumor (NT) tissue, with plectin perimembranous enrichment in tumor hepatocytes (*Figure 1C*; *Figure 1—figure supplement 1G*). Next, we compared plectin expression levels by immunoblotting in a panel of human HCC cell lines, which represent distinct stages of HCC development (*Boyault et al., 2007*). Consistent with mRNA and immunofluorescence analyses, poorly differentiated mesenchymal-like HCC cell lines (characterized by low E-cadherin and high vimentin levels) displayed elevated plectin levels, coinciding with higher migration speed when compared to well-differentiated HCC cell lines (*Figure 1D and E*).

To validate our findings in a well-established chemical carcinogen murine HCC model, we analyzed plectin expression in hepatic tumors formed 46 wk after diethylnitrosamine (DEN) injection in C57Bl/6J

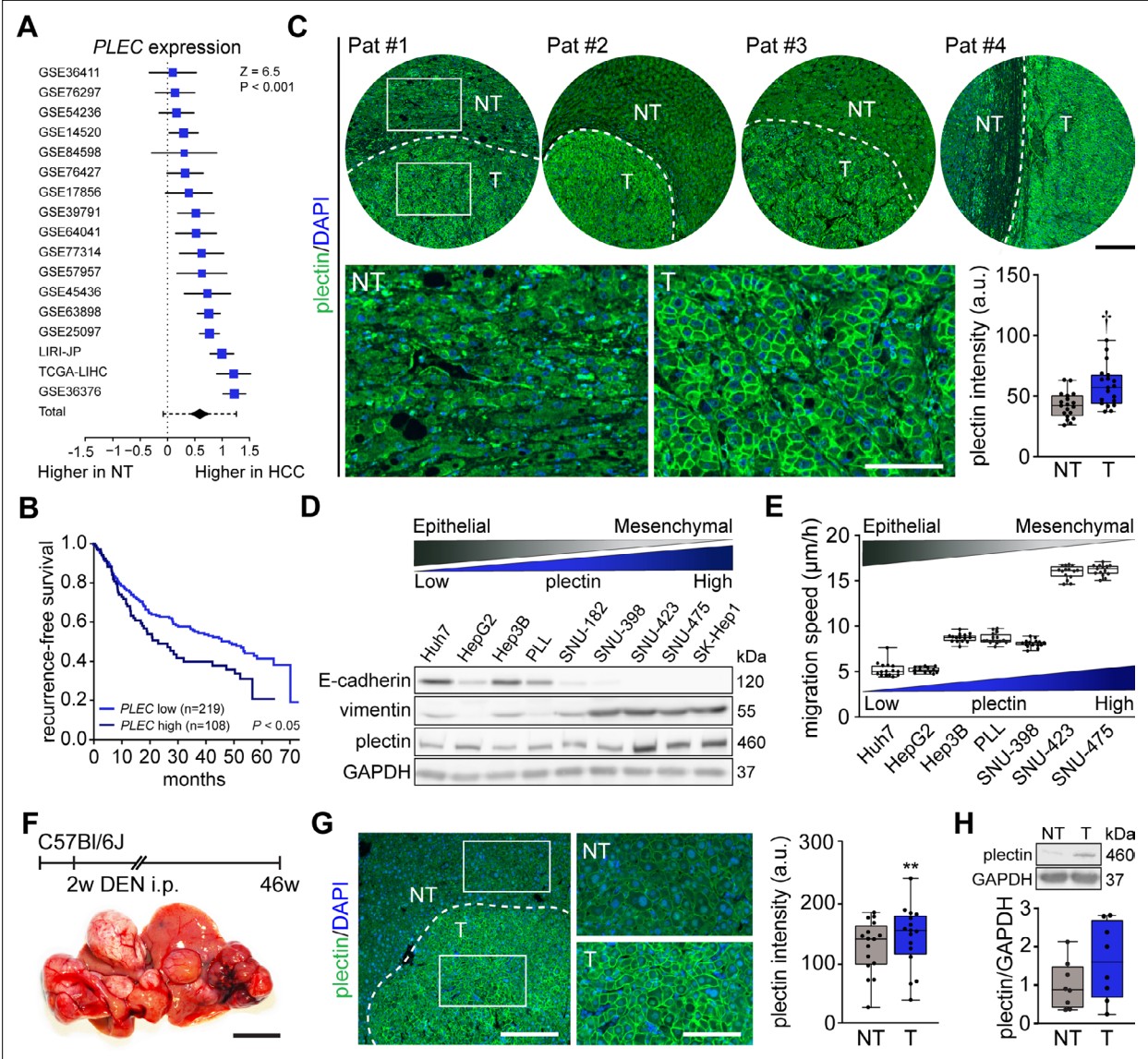

**Figure 1.** Plectin elevation in tumor hepatocytes is associated with hepatocellular carcinoma (HCC) progression and poor prognosis. (**A**) Meta-analysis of differential *plectin* (*PLEC*) mRNA expression in non-tumor (NT) liver and hepatocellular carcinoma (HCC) patients. Blue squares indicate the standardized mean difference (SMD) and 95% confidence interval of individual datasets. The black diamond shows the mean and 95% confidence interval for the combined SMD, while the whiskers indicate the 95% prediction interval. (**B**) Kaplan-Meier curve of recurrence-free survival of HCC patients with low *PLEC* (lower two tertiles, n=219) and high *PLEC* (top tertile, n=108) mRNA expression. Log-rank test; p<0.05. (**C**) Representative images of human HCC tissue sections immunolabeled for plectin (green). Nuclei, DAPI (blue). Dashed line, the borderline between non-tumor (NT) and tumor (T) area. Boxed areas, 4x images. Scale bars, 200 and 100 µm (boxed areas). Boxplot shows quantification of plectin fluorescence intensities in NT and T areas. The box represents the median, 25th, and 75th percentile; whiskers reach the last data point; dots, individual patients; N=19. Paired two-tailed *t*-test; †p<0.001. (**D**) Immunoblot analysis of indicated HCC cell lines with antibodies to plectin, E-cadherin, and vimentin. GAPDH, loading control. (**E**) Quantification of the speed of indicated HCC cell lines migrating in the scratch-wound assay. Boxplots show the median, 25th, and 75th percentile with whiskers reaching the last data point; dots, fields of view; n=15 (Huh7), 13 (HepG2), 15 (Hep3B), 15 (PLL), 15 (SNU-398), 15 (SNU-423), 15 (SNU-475) fields of view; N=3. (**F**) Hepatocarcinogenesis was induced in 2-wk-old C57Bl/6J mice by intraperitoneal injection of DEN. Representative image of the livers with multifocal HCC at 46 wk post-induction. Scale bar, 1 cm. (**G**) Representative image of DEN-induced HCC section immunolabeled for plectin (green). Nuclei, DAPI (blue). Dashed line, the borderline between non-tumor (NT) and tumor (T) area. Boxed areas, 2x images. Scale bars, 200 and 100 µm (boxed areas). Quantification of plectin fluorescence intensities in NT and T areas. Boxplot shows the median, 25th, and 75th percentile with whiskers reaching the last data point; dots, fields of view; n=16 fields of view; N=4. Paired two-tailed *t*-test; **p<0.01. (**H**) Immunoblot analysis of NT and T liver lysates. The boxplot shows relative plectin band intensities normalized to GAPDH. The box represents the median, 25th, and 75th percentile; whiskers reach the last data point; dots, individual mice; N=8.

The online version of this article includes the following source data and figure supplement(s) for figure 1:

*Figure 1 continued on next page*

Figure 1 continued

**Source data 1.** PDF file containing original western blots for *Figure 1D*, indicating the relevant bands.

**Source data 2.** Original files for western blot analysis displayed in *Figure 1D*.

**Source data 3.** PDF file containing original western blots for *Figure 1H*, indicating the relevant bands.

**Source data 4.** Original files for western blot analysis displayed in *Figure 1H*.

**Figure supplement 1.** Plectin is elevatted in hepatocellular carcinoma (HCC) across genders and etiologies.

mice (*Figure 1F*). Both quantitative immunofluorescence and immunoblot analyses indicated elevated plectin levels in T versus NT liver tissue (*Figure 1C, G and H*; *Figure 1—figure supplement 1H*). Moreover, enhanced plectin signal along hepatocyte membranes closely resembled the staining pattern found in patient HCC sections (*Figure 1C*), suggesting reliable translation from the human setting. Together, these results show that elevated plectin is associated with HCC progression both in human patients and animal models and indicates robust prognostic potential for patient survival.

## Plectin promotes hepatocarcinogenesis

To determine the functional consequences of plectin loss in liver tumor development, we analyzed the formation of DEN-induced HCCs in mice lacking plectin expression in the liver using magnetic resonance imaging (MRI). To achieve liver-specific plectin deletion, mice carrying a floxed plectin sequence (*Plec^fl/fl*) were crossed to mice expressing the Cre recombinase under the liver-specific albumin promoter (*Alb-Cre*). The resulting mice (*PlecΔAlb*) lack plectin expression in the liver (*Jirouskova et al., 2018*). Remarkably, MRI screening 32 and 44 wk post-injection revealed a significant reduction of tumor number and volume in *PlecΔAlb* mice compared to *Plec^fl/fl* controls (*Figure 2A–C*). Decreased tumor burden in the second cohort of *PlecΔAlb* mice was confirmed macroscopically 44 wk after DEN administration (*Figure 2D and E*). Notably, *PlecΔAlb* mice more frequently formed larger tumors, as reflected by overall tumor size increase (*Figure 2F*; *Figure 2—figure supplement 1A*), possibly implying reduced migration or increased cohesion of plectin-depleted cells (*Jirouskova et al., 2018*; *Xu et al., 2022*).

To address plectin's role in HCC at a cellular level, we genetically manipulated endogenous plectin in well-differentiated Huh7 and poorly differentiated SNU-475 human HCC cell lines (*Boyault et al., 2007*). Using the CRISPR/Cas-9 system we generated either knockouts (KO) or cells harboring endogenous plectin with deletion of the IF-binding domain (ΔIFBD) as functional knockouts (*Prechova et al., 2022*; *Figure 2—figure supplement 1B–D*). Gene editing was complemented by treatment with organoruthenium-based compound PST that inactivates plectin function (*Prechova et al., 2022*; *Meier et al., 2017*). If not stated otherwise, we applied PST in the final concentration of 8 µM, which corresponds to the 25% of $IC_{50}$ for Huh7 cells (*Figure 2—figure supplement 1E*). Consistent with the murine model, plectin inactivation resulted in a reduced number of Huh7 and SNU-475 colonies in a soft agar colony formation assay, with PST treatment closely mimicking the effect of genetic targeting (*Figure 2G*). Moreover, KO and ΔIFBD SNU-475 colonies were significantly smaller when compared to wild-type (WT) controls, with a similar trend observed for Huh7 cells (*Figure 2—figure supplement 1F*). Collectively, these data demonstrate the inhibitory effect of plectin inactivation on HCC progression in adhesion-independent conditions.

To further assess whether plectin is required for human HCC progression, we investigated the growth of subcutaneous Huh7 xenografts in immunodeficient NSG mice (*Figure 2H*; *Figure 2—figure supplement 1G*). Cells with disabled plectin developed significantly smaller tumors when compared with untreated WT cells (*Figure 2H*), mirroring the results of the colony-forming assay. The percentage of Ki67+ cells on immunolabeled xenograft sections, however, did not differ between experimental conditions (*Figure 2—figure supplement 1G*). These results show the reduced tumorigenic potential of human HCC cells when plectin is disabled either by CRISPR/Cas9-mediated gene ablation or pharmacologically with PST. Hence, by combining *in vivo* and *in vitro* approaches, we provide evidence that plectin promotes hepatocarcinogenesis.

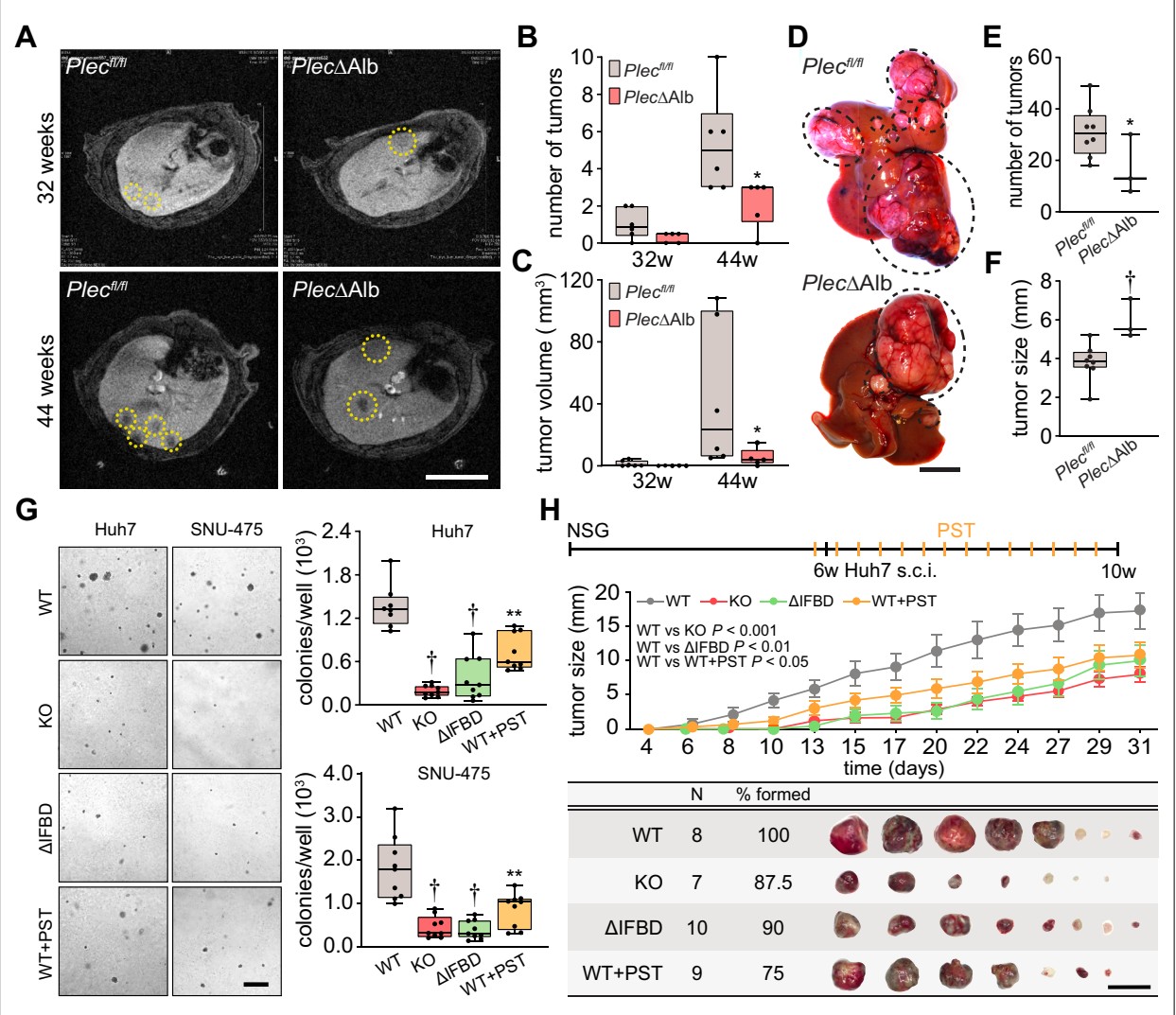

**Figure 2.** Plectin promotes hepatocellular carcinoma (HCC) growth. (**A**) Representative MRI images of *Plec^fl/fl* and *Plec*ΔAlb livers at 32 and 44 weeks post-diethylnitrosamine (DEN) injection. Dashed circles, tumors. Scale bar, 500 μm. (**B, C**) Quantification of tumor number (**B**) and volume (**C**) in *Plec^fl/fl* and *Plec*ΔAlb livers shown in (**A**). Boxplot shows the median, 25th, and 75th percentile with whiskers reaching the last data point; dots, individual mice; N=6 (*Plec^fl/fl*), 5 (*Plec*ΔAlb). Two-tailed *t*-test; *p<0.05. (**D**) Representative images of *Plec^fl/fl* and *Plec*ΔAlb livers at 44 wk post-induction. Dashed circles, tumors. Scale bar, 1 cm. (**E, F**) Quantification of the number (**E**) and size (**F**) of *Plec^fl/fl* and *Plec*ΔAlb tumors shown in (**D**). Boxplot shows the median, 25th, and 75th percentile with whiskers reaching the last data point; dots, individual mice; N=8 (*Plec^fl/fl*), 3 (*Plec*ΔAlb). Two-tailed *t*-test; *p<0.05; †p<0.001. (**G**) Representative images of colonies from WT, KO, ΔIFBD, and PST-treated WT (WT+PST) Huh7 and SNU-475 cells grown in soft agar. Scale bar, 500 μm. Boxplots show the number of Huh7 (upper graph) and SNU-475 (lower graph) cell colonies. The box represents the median, 25th, and 75th percentile with whiskers reaching the last data point; dots, agar wells; n=9 agar wells; N=3. Two-tailed *t*-test; **p<0.01; †p<0.001. (**H**) Six-week-old NSG mice were subcutaneously injected with indicated Huh7 cells into both hind flanks and were kept either untreated (WT, KO, and ΔIFBD) or bidiurnally treated by orogastric gavage of plecstatin (WT+PST) as indicated in the upper bar. Mice were sacrificed 4 wk post-injection and xenografts were dissected. The graph shows the time course of xenograft growth. Data are shown as mean ± SEM; n=8 (WT), 7 (KO), 10 (ΔIFBD) and 9 (WT+PST) tumors; N=4 (WT), 4 (KO), 5 (ΔIFBD) and 6 (WT+PST). Two-way ANOVA. The table shows the number (N), percentage, and representative images of formed xenografts. Scale bar, 2 cm.

The online version of this article includes the following source data and figure supplement(s) for figure 2:

**Figure supplement 1.** Plectin influences hepatocellular carcinoma (HCC) progression *in vivo*.

**Figure supplement 1—source data 1.** PDF file containing original western blots for *Figure 2—figure supplement 1C*, indicating the relevant bands.

**Figure supplement 1—source data 2.** Orignal files for western blot analysis displayed in *Figure 2—figure supplement 1C*.

## Plectin controls oncogenic FAK, MAPK/Erk, and PI3K/Akt signaling in HCC cells

To identify potential molecular effectors and signaling pathways mediating the tumor suppressive effects of plectin inactivation, we profiled the proteomes of WT, KO, and PST-treated WT SNU-475 cells using MS-based shotgun proteomics and phosphoproteomics (*Figure 3A–C*; *Figure 3—figure supplement 1A and B*). Using a label-free quantification strategy, a total of 5440 protein groups and 3573 phosphosites were detected. We found 265 protein groups significantly regulated (FDR <0.05; s0=0.01) upon plectin ablation when comparing WT and KO SNU-475 proteomes (*Figure 3B*). Ingenuity Pathway Analysis (IPA) revealed major plectin-dependent regulation of signaling pathways related to the actin cytoskeleton, such as 'RhoA signaling', 'Actin cytoskeleton signaling', 'Integrin signaling', and 'Signaling by Rho family GTPases' (*Figure 3B*). Similarly, 313 regulated phosphosites indicated a major impact on actin, as well as 'ILK signaling', 'FAK signaling', and 'Molecular mechanisms of cancer' among the most altered pathways (*Figure 3B*).

Analysis of proteome differences between WT and PST-treated cells identified abundance changes (FDR <0.05; s0=0.01) in 1214 proteins and 326 phosphoproteins (*Figure 3C*). A comparison of KO and PST signatures using IPA revealed an overlap of 90 proteins and 61 phosphosites (*Figure 3—figure supplement 1A*). Consistently, the IPA annotation linked also PST signature to integrin- and cytoskeleton-related signaling pathways such as 'ILK signaling', 'Integrin signaling', 'RhoA signaling', and 'Actin cytoskeleton signaling' (*Figure 3C*). Taken together, our proteomic analyses suggest a regulatory role for plectin in the mechanosensitive, cell adhesion-linked signaling which is critical for cancer development and dissemination (*Cooper and Giancotti, 2019*; *Hoxhaj and Manning, 2020*; *Sun et al., 2016*).

To independently confirm our MS findings, we performed extensive immunoblot analysis of WT, KO, ΔIFBD, and PST-treated Huh7 and SNU-475 cells with a focus on integrin-associated adhesome network (*Figure 3D*; *Figure 3—figure supplement 1C*). In agreement with our proteomic analyses, plectin inactivation resulted in considerable changes in expression levels of integrin adhesion receptors (integrins αV and β1) as well as other FA constituents (i.e. talin, vinculin, and paxillin). Moreover, immunoblotting revealed in cells with disabled plectin either generally altered expression (FAK, Akt, Erk1/2, ILK, and PI3K) and/or reduced phosphorylation (Akt, Erk1/2, and PI3K) of key effectors downstream of integrin-mediated adhesion. Although these alterations were not found systematically in both cell lines and condition (reflecting thus presumably their distinct differentiation grade and plectin inactivation efficacy), collectively these data confirmed plectin-dependent adhesome remodeling together with attenuation of oncogenic FAK, MAPK/Erk, and PI3K/Akt pathways upon plectin inactivation (*Figure 3E*).

## Plectin-dependent disruption of cytoarchitecture accounts for hampered migration of HCC cells

As plectin acts as a major organizer of cytoskeletal networks (*Prechova et al., 2023*), we next investigated cytoskeletal organization in HCC cells by immunofluorescence microscopy. To circumvent considerable variability in cellular morphology, which largely obscures quantitative assessment of cytoarchitecture, we seeded WT, KO, ΔIFBD, and PST-treated SNU-475 cells on crossbow-shaped micropatterns (*Jiu et al., 2015*). Reminiscent of plectin-deficient fibroblasts (*Gregor et al., 2014*; *Burgstaller et al., 2010*), plectin inactivation in SNU-475 cells produced less delicate vimentin networks compared to WT cells, with filaments often bundled and sometimes collapsing into vimentin clumps (*Figure 4A and B*). A quantitative analysis revealed uneven distribution of vimentin filaments throughout the cytoplasm of KO, ΔIFBD, and PST-treated WT cells as evidenced by the distance between the position of the center of vimentin intensity mass and the cell center (*Figure 4—figure supplement 1A–C*). In addition to the aberrant vimentin phenotype, we noticed a dramatic reduction in longitudinal dorsal actin stress fibers and transversal arcs, as well as pronounced ventral stress fibers in plectin-disabled cells (*Figure 4A, C and D*). Moreover, we detected a reduction in F-actin fluorescence intensity in both Huh7 and SNU-475 KO cells, as well as a decrease of atomic force microscopy (AFM)-inferred cellular stiffness as a functional readout for a well-formed cytoskeleton (*Na et al., 2009*; *Figure 4—figure supplement 1D–G*).

Given the extent of plectin-dependent adhesome remodeling (*Figure 3D and E*; *Figure 3—figure supplement 1C*), we next assessed whether plectin inactivation affects the morphology and

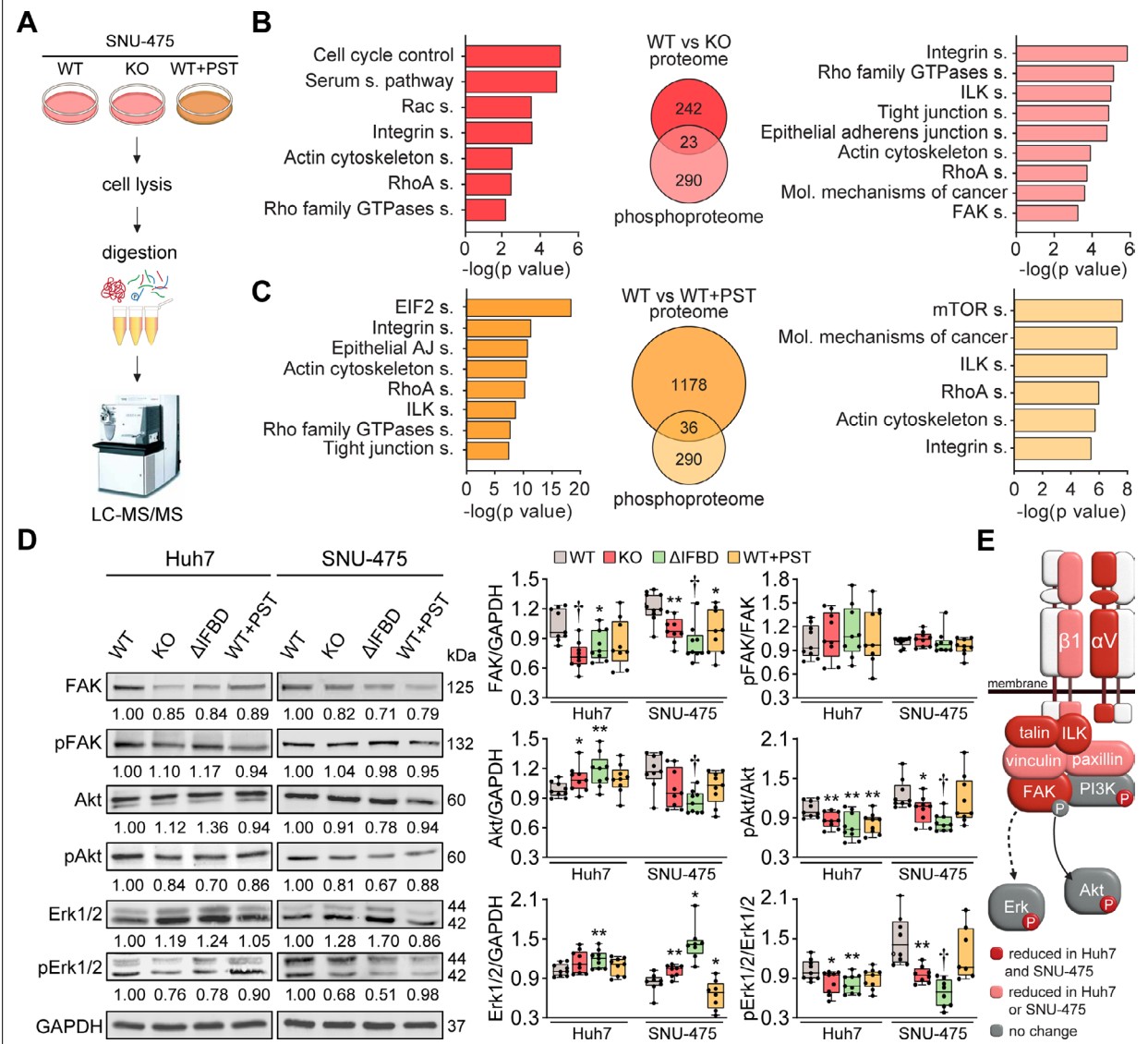

**Figure 3.** CRISPR/Cas9- or Plecstatin-1 (PST)-mediated plectin inactivation attenuates hepatocellular carcinoma (HCC) oncogenic potential through FAK, Erk1/2, and PI3K/Akt axis. (**A**) Schematic of MS-based proteomic analysis of wild-type (WT), knockout (KO), and PST-treated WT (WT+PST) SNU-475 cells. (**B, C**) Ingenuity Pathway Analysis (IPA) canonical signaling pathways predicted from differentially expressed proteins identified by proteomics (left) and phosphoproteomics (right) in WT vs. KO (**B**) and WT vs. WT+PST (**C**) proteomes. Venn diagrams show relative proportions of differentially expressed proteins. Two-sided Student´s t-test with multiple testing correction: FDR < 0.05; s0 = 0.1; triplicates. (**D**) Quantification of FAK, phospho-Tyr397-FAK (pFAK), Akt, phospho-Ser473-Akt (pAkt), Erk1/2, and phospho-Thr202/Tyr204-Erk (pErk) in indicated Huh7 and SNU-475 cell lines by immunoblotting. GAPDH, loading control. The numbers below lines indicate relative band intensities normalized to average WT values. Boxplots show relative band intensities normalized to GAPDH or non-phosphorylated protein. The box represents the median, 25th, and 75th percentile with whiskers reaching the last data point; dots, individual experiments; N=9. Two-tailed t-test; *p<0.05; **p<0.01; †p<0.001. (**E**) Schematic representation of immunoblot analyses of adhesome-associated signaling shown in (**D**) and (Extended Data *Figure 3—figure supplement 1C*). Proteins with significantly reduced expression levels and/or phosphorylation status (P) upon plectin inactivation in both HCC cell lines are highlighted in red, proteins with significantly reduced expression levels upon plectin inactivation in either Huh7 or SNU-475 cells are highlighted in pink.

The online version of this article includes the following source data and figure supplement(s) for figure 3:

**Source data 1.** PDF file containing original western blots for *Figure 3D*, indicating the relevant bands.

**Source data 2.** Original files for western blot analysis displayed in *Figure 3D*, *Figure 3—figure supplement 1C*.

**Source data 3.** Original files for western blot analysis displayed in *Figure 3D*, *Figure 3—figure supplement 1C*.

**Figure supplement 1.** Integrin-associated signaling is altered in SNU-475 cells upon CRISPR/Cas9- or Plecstatin-1 (PST)-mediated plectin inactivation.

**Figure supplement 1—source data 1.** PDF file containing original western blots for *Figure 3—figure supplement 1C*, indicating the relevant bands.

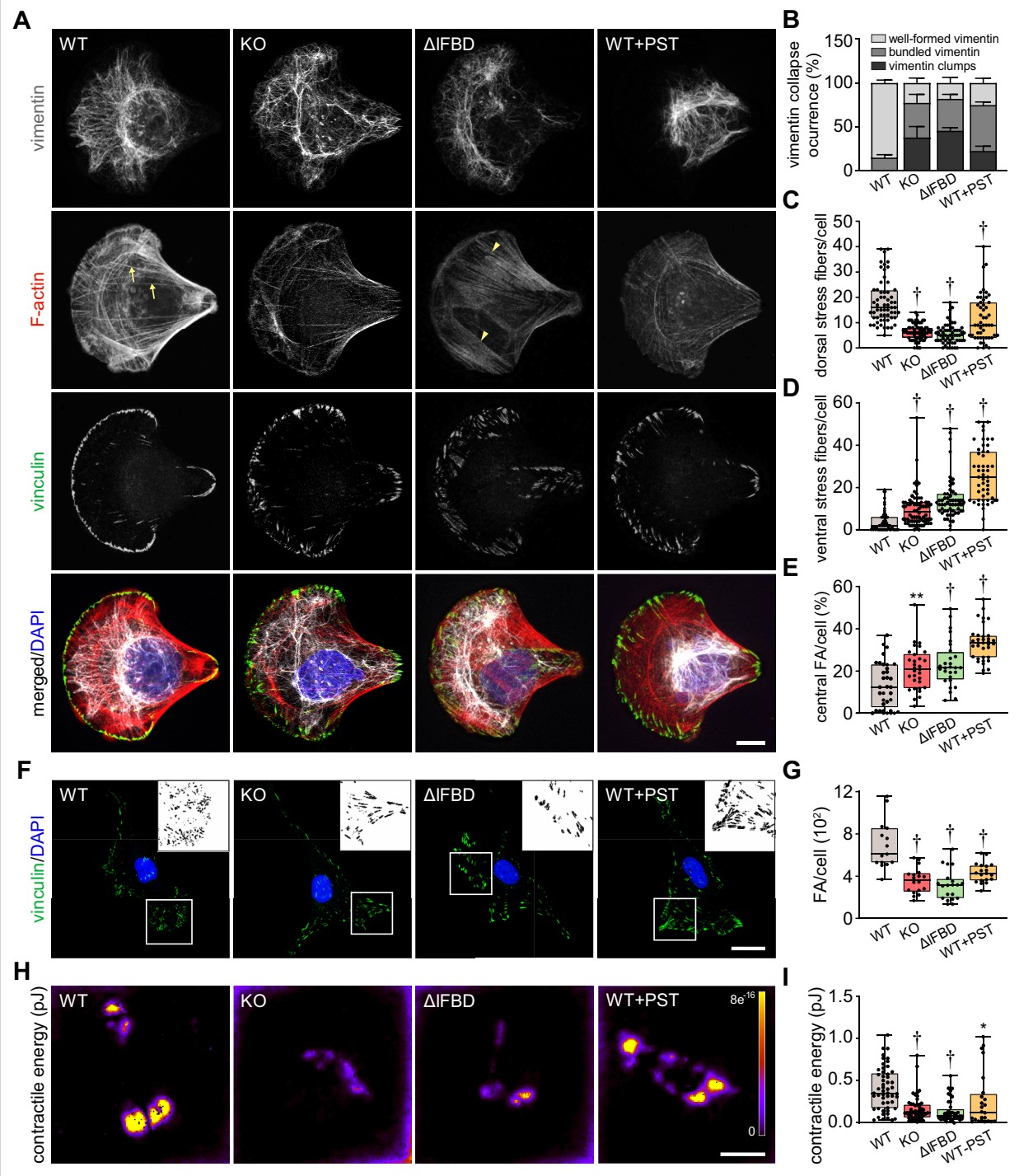

**Figure 4.** Disruption of cytoskeletal networks upon plectin inactivation accounts for reduced contractility and aberrant adhesions in hepatocellular carcinoma (HCC) cells. (**A**) Representative confocal images of crossbow-shaped fibronectin micropattern-seeded wild-type (WT), knockout (KO), ΔIFBD, and Plecstatin-1 (PST)-treated WT (WT+PST) SNU-475 cells stained for F-actin (red), vinculin (green), and vimentin (gray). Nuclei, DAPI (blue). Arrows, dorsal stress fibers; arrowheads, ventral stress fibers. Scale bar, 10 μm. (**B**) Quantification of the percentage of cells (shown in (**A**)) with well-formed, bundled, and clump-containing vimentin networks. Data are shown as mean ± SEM; n=60 (WT), 68 (KO), 55 (ΔIFBD), 50 (WT+PST) cells; N=4 (WT, KO, IFBD), 3 (WT+PST). (**C, D**) Quantification of the number of dorsal (**C**) and ventral (**D**) actin stress fibers in cells shown in (**A**). Boxplots show the median, 25th, and 75th percentile with whiskers reaching the last data point; dots, individual cells; n=60 (WT), 68 (KO), 55 (ΔIFBD), 50 (WT+PST); N=4 (WT, KO, IFBD), 3 (WT+PST). Two-tailed *t*-test; †p<0.001. (**E**) Quantification of focal adhesions (FAs) located within the interior of cells (central) shown in (**A**). Boxplot shows the median, 25th, and 75th percentile with whiskers reaching the last data point; dots, individual cells; n=25 (WT), 26 (KO), 23

*Figure 4 continued on next page*

*Figure 4 continued*

(ΔIFBD), 28 (WT+PST); N=3. **p<0.01; †p<0.001. (**F**) Representative confocal images of WT, KO, ΔIFBD, and PST-treated WT (WT+PST) SNU-475 cells immunolabeled for vinculin (green). Nuclei, DAPI (blue). Boxed areas, and representative FA clusters shown as segmented binary maps in 2x enlarged insets. Scale bar, 30 μm. (**G**) Quantification of FA number in cells shown in (**F**). Boxplot shows the median, 25th, and 75th percentile with whiskers reaching the last data point; dots, individual cells; n=15 (WT), 18 (KO), 20 (ΔIFBD), 19 (WT+PST); N=3. Two-tailed *t*-test; †p<0.001. (**H**) Pseudocolor spatial maps of contractile energy determined by TFM in WT, KO, ΔIFBD, and PST-treated WT (WT+PST) SNU-475 cells. Scale bar, 50 μm. (**I**) Quantification of contractile energy in cells shown in (**H**). Boxplots show the median, 25th, and 75th percentile with whiskers reaching the last data point; dots, individual cells; n=54 (WT), 53 (KO), 41 (ΔIFBD), 24 (WT+PST) cells; N=4. Two-tailed *t* test; *p<0.05; **p<0.01; †p<0.001.

The online version of this article includes the following figure supplement(s) for figure 4:

**Figure supplement 1.** Cytoskeletal networks and cell stiffness are altered upon plectin inactivation.

localization of FAs in vinculin-immunolabeled SNU-475 cells. Remarkably, while FAs of micropattern-seeded WT cells were mostly located at the cell periphery, FAs of plectin-disabled cells were frequently found within the cell interior (*Figure 4A and E*). Moreover, plectin inactivation resulted in an overall reduced number of FAs, and the FAs that remained were larger and more elongated than in WT cells (*Figure 4F and G*; *Figure 4—figure supplement 1H and I*). To test whether the changes in actin/FA configuration affected adhesion-transmitted forces, we performed traction force microscopy (TFM; *Figure 4H and I*). The smaller FAs found in WT cells transmitted significantly higher contractile energy than KO, ΔIFBD, and PST-treated cells, indicating that FAs in plectin-deficient cells were less functional than in WT.

Functional transmission of actomyosin-generated forces across FAs constitutes a prerequisite for cellular locomotion (*Bodor et al., 2020*). Therefore, we examined the effect of plectin inactivation on the migration of HCC cells. As anticipated, both Huh7 and SNU-475 cells exhibited a decrease in migration speed upon plectin targeting in the scratch wound healing assay (*Figure 5A and B*; *Figure 5—figure supplement 1A*). It is noteworthy that migrating plectin-disabled SNU-475 cells exhibited more cohesive, epithelial-like features while progressing collectively. By contrast, WT SNU-475 leader cells were more polarized and found to migrate into scratch areas more frequently than their plectin-deficient counterparts (*Figure 5—figure supplement 1B*). Consistent with this observation, individually seeded SNU-475 cells less frequently assumed a polarized, mesenchymal-like shape upon plectin inactivation in both 2D and 3D environments (*Figure 5C*). Moreover, plectin-inactivated SNU-475 cells exhibited a decrease in N-cadherin and vimentin levels when compared to WT counterparts (*Figure 5—figure supplement 1C*).

In addition to slower general migration, we also found the epithelial growth factor (EGF)-guided migration potential of individual KO, ΔIFBD, and PST-treated cells to be significantly reduced compared to WT cells. Consistent with previous findings (*Gregor et al., 2014*), plectin-disabled cells traversed less linear trajectories in both random and directed scenarios (*Figure 5D and E*; *Figure 5—figure supplement 1D and E*). To determine whether plectin is involved in migration-associated cellular shape dynamics, we further investigated protrusions of SNU-475 cells using morphody-namic contour analysis (*Yolland et al., 2019*). Our analysis revealed a higher protrusion frequency of randomly migrating WT compared to plectin-disabled cells (*Figure 5—figure supplement 1F*), while no differences in protrusion orientation were observed (*Figure 5F–H*). In sharp contrast, plectin ablation dramatically reduced the capacity of KO and ΔIFBD cells to form stable protrusions in the direction of chemotactic motion (*Figure 5F–H*), although only a marginal effect on the protrusivity was observed (*Figure 5—figure supplement 1F*). Collectively, these results show that plectin is essential for the proper cytoskeletal configuration of HCC cells and their cytoskeleton-linked FAs. Moreover, they provide evidence that aberrant cytoarchitecture of plectin-disabled cells accounts for the failure to effectively exert traction forces and actively reconfigure body shape, both of which are required for HCC cell migration.

## Plectin inactivation reduces HCC cell invasion and lung colonization

To investigate whether disruption of cytoarchitecture in plectin-disabled HCC cells also affected 3D migratory behavior, we compared the activity of WT, KO, ΔIFBD, and PST-treated SNU-475 cells in transwell and spheroid invasion assays. In both assays, plectin inactivation significantly reduced inva-sion potential compared to WT cells (*Figure 6A–C*; *Figure 6—figure supplement 1A*). Unexpectedly, plectin-targeted cells also degraded dramatically less FITC-labeled gelatin, suggesting that slower

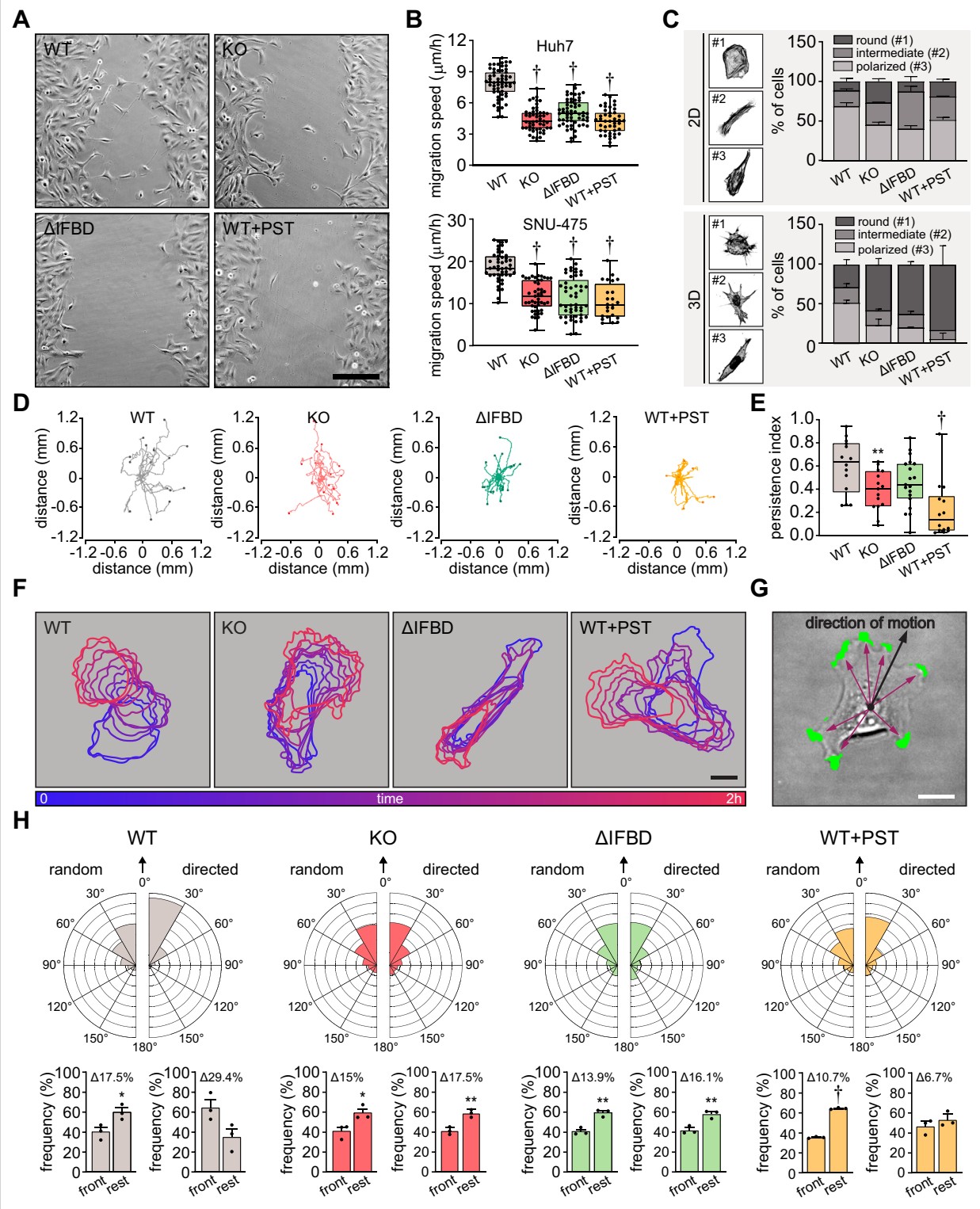

**Figure 5.** Plectin links the migration potential of hepatocellular carcinoma (HCC) cells to cell shape dynamics. (**A**) Representative phase contrast images of wild-type (WT), knockout (KO), ΔIFBD, and PST-treated WT (WT+PST) SNU-475 cells migrating in the scratch-wound assay for 14 hr. Note individual, highly polarized WT cells frequently migrate into scratch areas. Scale bar, 200 µm. (**B**) Quantification of migration speed of indicated Huh7 (upper graph) and SNU-475 (lower graph) cells. Boxplots show the median, 25th, and 75th percentile with whiskers reaching the last data point; dots, fields of view; *n* (Huh7)=59 (WT), 51 (KO), 58 (ΔIFBD), 43 (WT+PST); *n* (SNU-475)=47 (WT), 47 (KO), 50 (ΔIFBD), 24 (WT+PST); *N* (Huh7)=3; *N* (SNU-475)=5 (WT, KO, ΔIFBD), 3 (WT+PST). Two-tailed *t*-test; † p<0.001. (**C**) Representative confocal images of F-actin stained WT, KO, ΔIFBD, and PST-treated WT

*Figure 5 continued on next page*

*Figure 5 continued*

(WT+PST) SNU-475 cells grown on fibronectin-coated coverslips (2D) or in collagen (3D) and classified as round (#1), intermediate (#2), and polarized (#3) shape. Quantification of the percentage of cell shape categories in indicated 2D and 3D SNU-475 cell cultures. Data are shown as mean ± SEM; *N* (2D)=3; *N* (3D)=5 (WT), 3 (KO, ΔIFBD), 2 (WT+PST). (**D**) Spider plots with migration trajectories of WT, KO, ΔIFBD, and PST-treated WT (WT+PST) SNU-475 cells tracked during 16 hr of EGF-guided migration; dots, the final position of each single tracked cell. (**E**) Quantification of processivity indices of WT, KO, ΔIFBD, and PST-treated WT (WT+PST) SNU-475 cells shown in (**D**). Boxplot shows the median, 25th, and 75th percentile with whiskers reaching the last data point; dots, individual cells; n=15 (WT), 15 (KO), 19 (ΔIFBD), 14 (WT+PST); N=3. Two-tailed *t*-test; **p<0.01;† p<0.001. (**F**) Representative time sequences of the WT, KO, ΔIFBD, and Plecstatin-1 (PST)-treated WT (WT+PST) SNU-475 cell contours during EGF-guided migration. Color coding indicates the time of cell position acquired in 10 min intervals. Scale bar, 20 µm. (**G**) Representative phase contrast image of SNU-475 cell with protrusions (green) segmented from superimposed contours used in morphodynamic analysis. Extension vectors (purple arrows) were drawn from the center of the cell nucleus towards individual protrusions and related to the direction of cell motion (black arrow). Scale bar, 20 µm. (**H**) Rose graphs show the percentage of extension vector directions in 30° cones, normalized to the directions of random and EGF-guided (directed) motions (0°; arrows) of WT, KO, ΔIFBD, and PST-treated WT (WT+PST) SNU-475 cells. n=9752 extensions in 22 cells (WT random), 4167 extensions in 15 cells (WT directed), 8394 extensions in 19 cells (KO random), 5107 extensions in 15 cells (KO directed), 8362 extensions in 21 cells (ΔIFBD random), 5809 extensions in 19 cells (ΔIFBD directed), 9450 extensions in 20 cells (WT+PST random), 4350 extensions in 14 cells (WT+PST directed); N=3. Bar graphs show the percentage of cell extensions formed either in the direction of motion (frontal, 30° to −30° cones) or along the rest of the cell perimeter (rest). Data are shown as mean ± SEM; dots, biological replicates; N=3. Two-tailed *t*-test; *p<0.05; **p<0.01;† p<0.001.

The online version of this article includes the following source data and figure supplement(s) for figure 5:

**Figure supplement 1.** Plectin inactivation impairs the migration of hepatocellular carcinoma (HCC) cells.

**Figure supplement 1—source data 1.** PDF file containing original western blots for *Figure 5—figure supplement 1C*, indicating the relevant bands.

**Figure supplement 1—source data 2.** Original files for western blot analysis displayed in *Figure 5—figure supplement 1C*.

invasion is accompanied by defects in ECM degradation (*Figure 6D*; *Figure 6—figure supplement 1B*).

To monitor plectin effects on shape dynamics in a 3D environment, we recorded WT and KO SNU-475 cells by time-lapse video microscopy in a matrigel invasion assay. Invading WT cells exhibited polarized protrusions followed by cell body displacement in the direction of the nascent protrusion (*Figure 6—video 1*; *Figure 6E*). By contrast, randomly oriented thinner protrusions of KO cells were often retracted shortly after formation. Markedly thinner and branched KO protrusions were confirmed by subsequent immunofluorescence microscopy (*Figure 6E*). Similar to what we observed in the 2D assay, KO cells failed to invade in the direction of these transient protrusions (*Figure 6—video 1*; *Figure 6E*). Hence, plectin-controlled cytoarchitecture facilitates both 2D and 3D HCC cell migration.

Tumor, node, metastasis (TNM) classification of an HCC meta-cohort with clinically annotated tumors from HCC patients (n=978) demonstrated that high *PLEC* mRNA expression is associated with advanced TNM stages (*Figure 6F*). To elucidate the impact of plectin inactivation on HCC dissemination, we conducted the lung colonization assay using both Huh7 and SNU-475 cells (*Figure 6G–I*; *Figure 6—figure supplement 1C–F*). To this end, we administered red firefly luciferase and GFP (RedFLuc-GFP)-expressing WT and KO cells intravenously in 5-week-old NSG mice. Whereas mice receiving WT Huh7 (but not SNU-475; data not shown) cells succumbed rapidly to disease, mice receiving KO cells exhibited prolonged survival (*Figure 6G*). To identify the early phase of metastasis formation, we next monitored the HCC cell retention in the lungs using *in vivo* bioluminescence imaging (*Figure 6H*). This experimental cohort was expanded for WT-injected mice which were administered PST bidiurnally for 5 wk (WT+PST). Mice were sacrificed 5 wk post-injection when the first luminescence-positive chest areas were detected (*Figure 6H*) and cleared whole lung lobes were analyzed by lattice light sheet fluorescence microscopy (*Figure 6I*). Although no macroscopic Huh7 nodules were visible, we found a prominent reduction in the number and volume of GFP-positive KO- and WT+PST-derived metastatic nodules. Thus, both CRISPR/Cas9-based and pharmacological plectin inactivation in HCC potently inhibits metastatic load in the lungs, identifying plectin as a potential target against tumor dissemination *in vivo*.

## Genetic and pharmacological plectin targeting prevents hepatocarcinogenesis through signatures shared by animal models and patients

To further investigate the translational potential of PST treatment, we evaluated the effects of PST administration on hepatocarcinogenesis in the additional murine model. To this end, we induced

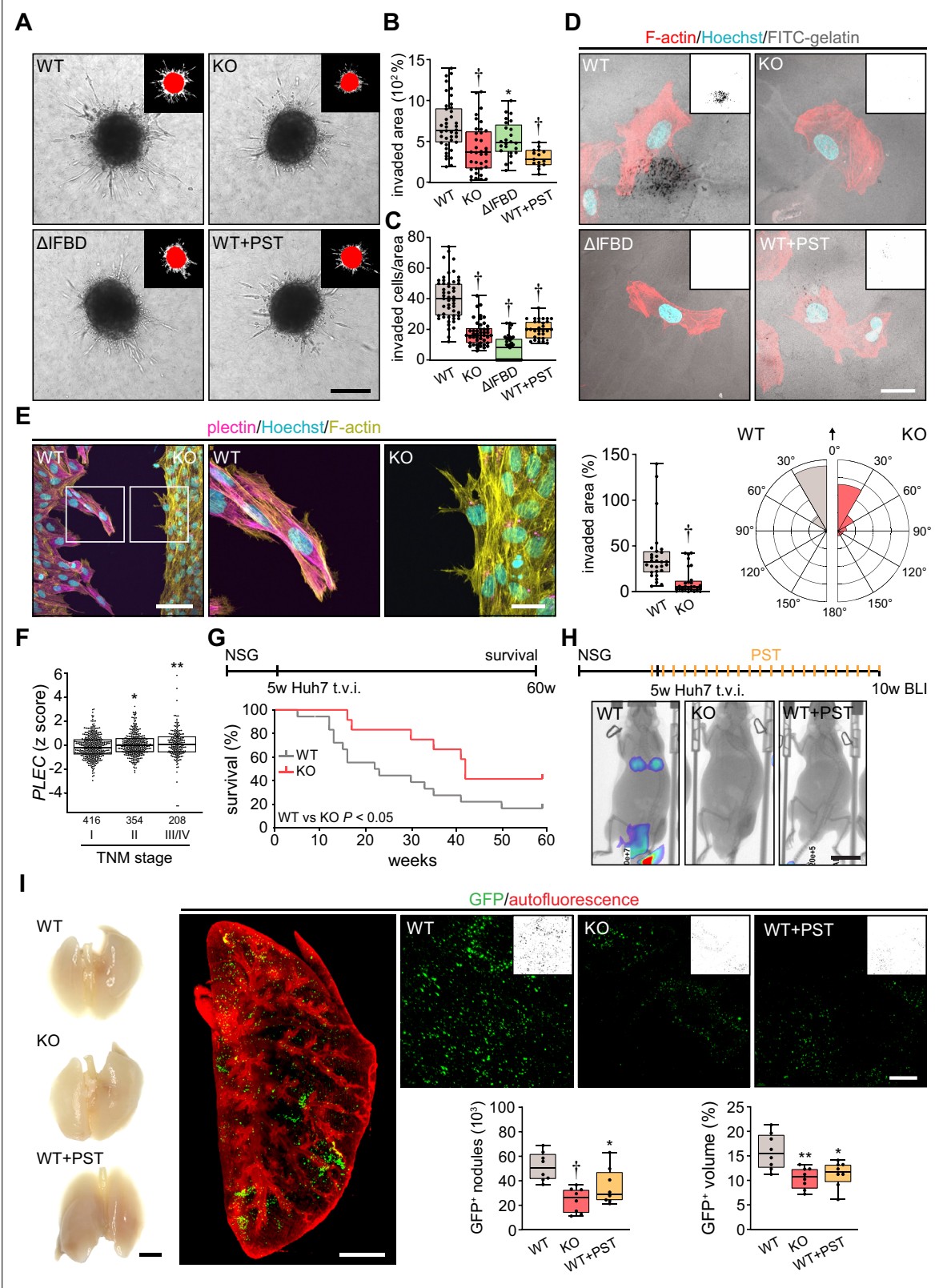

**Figure 6.** Plectin inactivation inhibits hepatocellular carcinoma (HCC) invasion and metastasis. (**A**) Representative images of wild-type (WT), knockout (KO), ΔIFBD, and Plecstatin-1 (PST)-treated WT (WT+PST) SNU-475 spheroids grown for 3 d in collagen mixture. Insets, superimposed binary masks of initial (red) and final (white) spheroid area. Scale bar, 200 μm. (**B**) Quantification of the invaded area calculated as the percentage of the initial spheroid area from day 0. Boxplots show the median, 25th, and 75th percentile with whiskers reaching the last data point; dots, individual spheroids;

*Figure 6 continued on next page*

*Figure 6 continued*

n=47 (WT), 44 (KO), 34 (ΔIFBD), 25 (WT+PST) spheroids; N=5 (WT, KO), 4 (ΔIFBD), 3 (WT+PST). Two-tailed *t*-test; **p<0.01; †p<0.001. (**C**) Quantification of the number of indicated cells invaded in Matrigel transwell assay. Boxplots show the median, 25th, and 75th percentile with whiskers reaching the last data point; dots, fields of view; n=51 (WT), 45 (KO), 38 (ΔIFBD), 31 (WT+PST) fields of view; N=4 (WT, KO), 3 (ΔIFBD, WT+PST). Two-tailed *t*-test; †p<0.001. (**D**) Representative confocal micrographs of WT, KO, ΔIFBD, and PST-treated WT (WT+PST) SNU-475 cells grown on FITC-labeled gelatin (gray) for 24 hr and stained for F-actin (red). Nuclei, Hoechst (blue). Insets, segmented binary masks of FITC-gelatin signal. Black regions correspond to gelatin areas degraded by individual cells. Scale bar, 30 µm. (**E**) Representative confocal images of WT and KO SNU-475 cells during the Matrigel invasion assay, stained for plectin (magenta) and F-actin (yellow). Nuclei, Hoechst (blue). See *Figure 6—video 1*. Boxed areas, 3x images. Scale bars, 100 and 30 µm (boxed areas). Boxplot shows the invaded area calculated as the percentage of the initial area covered by WT and KO cells. The box represents the median, 25th, and 75th percentile with whiskers reaching the last data point; dots, fields of view; n=29 fields of view; N=2. Rose graphs show the percentage of extension vector directions in 30° cones, normalized to the directions of cell motions (0°; arrow) during matrigel invasion. n=857 extensions in 18 cells (WT), 623 extensions in 12 cells (KO); N=2. Two-tailed *t*-test; †p<0.001. (**F**) Relative *plectin* (*PLEC*) mRNA expression in samples collected from HCC patient meta-cohort clustered based on tumor, node, metastasis (TNM) classification (stage I-IV). The meta-cohort includes 6 different datasets from five platforms (for details, see Materials and methods section). The numbers of participants per stage are indicated in the graph. Scattered boxplots show individual data points, median, 25th, and 75th percentile; N=978. Wilcoxon rank-sum test; *p<0.05; **p<0.01. (**G**) The 5-wk-old NSG mice were injected into tail vein (tail vein injection; t.v.i.) with WT and KO RedFLuc-GFP-expressing Huh7 cells generated for lung colonization assay. Kaplan-Meier curves show the overall survival of mice injected with the cells indicated. N = 14 (WT), 13 (KO). Long-rank test, p<0.05. (**H**) The 5-week-old NSG mice were injected (t.v.i.) with indicated RedFLuc-GFP-expressing Huh7 cells. WT cell-bearing mice were kept either untreated or every second day provided with orogastric gavage of plecstatin (WT+PST) as indicated. Five weeks post-injection mice were screened by whole-body bioluminescence imaging (BLI). Representative BLI images of WT, KO, and PST-treated WT (WT+PST) Huh7 cells-bearing mice are shown. Scale bar, 2 cm. (**I**) Representative images of lungs dissected from mice shown in (**H**). Scale bar, 1 cm. Representative lattice light sheet fluorescence image of clear, unobstructed brain imaging cocktails (CUBIC)-cleared lung lobe immunolabeled with antibodies against GFP (green). Autofluorescence visualizing the lobe structures is shown in red. Scale bar, 2 mm. Representative magnified images from lung lobes with GFP-positive WT, KO, and WT+PST Huh7 nodules. Insets, segmented binary masks of GFP-positive metastatic nodules. Scale bar, 400 µm. Boxplots show metastatic load in the lungs expressed as the number (left graph) and relative volume (right graph) of indicated GFP-positive (GFP+) nodules. The box represents the median, 25th, and 75th percentile with whiskers reaching the last data point; dots, lung lobes; n=8 lung lobes; N=4. Two-tailed *t*-test; *p<0.05; **p<0.01; †p<0.001.

The online version of this article includes the following video and figure supplement(s) for figure 6:

**Figure supplement 1.** Plectin inactivation reduces the invasiveness of hepatocellular carcinoma (HCC) cells.

**Figure 6—video 1.** Representative video of wild-type (WT) and knockout (KO) SNU-475 cells invading the matrigel.

https://elifesciences.org/articles/102205/figures#fig6video1

Time-lapse covers total 21 hr with frame taken every 15 min (~15 min elapsed time per frame of the movie). Scale bar, 200 µm. The fixed and immunolabeled cells from the endpoint of this experiment are shown in *Figure 6E*.

---

multifocal HCC tumors by hydrodynamic delivery of a *MYC* (Myc)-encoding element together with a CRISPR/Cas9 construct targeting *Tp53* (sgTp53; *Revia et al., 2022*). To test whether HCC onset and progression are sensitive to pharmacological targeting of plectin, we monitored Myc;sgTp53-driven tumor development in *Plec^{fl/fl}*, *PlecΔAlb*, and PST-treated *Plec^{fl/fl}* male mice using MRI (*Figure 7A*). Consistent with our *in vitro* observations, MRI analysis at 4, 6, and 9 wk post-induction revealed that both genetic and pharmacological plectin inactivation results in a substantial reduction in the average tumor number per mouse and the tumor incidence (*Figure 7A*). Stalled development of *PlecΔAlb* and PST-treated *Plec^{fl/fl}* tumors was also reflected by a decrease in liver/body weight ratio in another male cohort sacrificed at 6 wk post-induction (*Figure 7B and C*). The quantitative immunofluorescence microscopy revealed comparable rates of proliferation and apoptosis in Myc;sgTp53-induced tumors across experimental conditions (*Figure 7—figure supplement 1A and B*). Comparable trends in liver/body weight ratio and tumor incidence were also found in a female cohort sacrificed 8 wk post-induction (*Figure 7—figure supplement 1C and D*).

To better understand the antitumor effects observed in PST-treated mice, we performed proteomics on Myc;sgTp53-treated *Plec^{fl/fl}*, *PlecΔAlb*, and PST-treated *Plec^{fl/fl}* livers. Consistent with (phospho) proteomic and immunoblot analyses of HCC cell lines (*Figure 3A–E*) we found a high level of similarity between *PlecΔAlb* and PST-treated *Plec^{fl/fl}* signatures (*Figure 7D*; *Figure 7—figure supplement 2A and B*). In addition, gene set enrichment analysis (GSEA; *Subramanian et al., 2005*) revealed enrichment in 'PI3K/Akt' or 'Hippo/YAP signaling' pathways (*Figure 7D*). Although the data from liver tissue proteomics showed some degree of variation, enrichment of tension-dependent signatures points toward similar trends found in *in vitro* scenarios. To further translate our findings to the human setting, we correlated plectin transcript levels with differentially expressed signatures identified in proteomic analysis of HCC cells (*Figure 3B–D*). Through analysis of 1268 HCC patients, we found gene sets

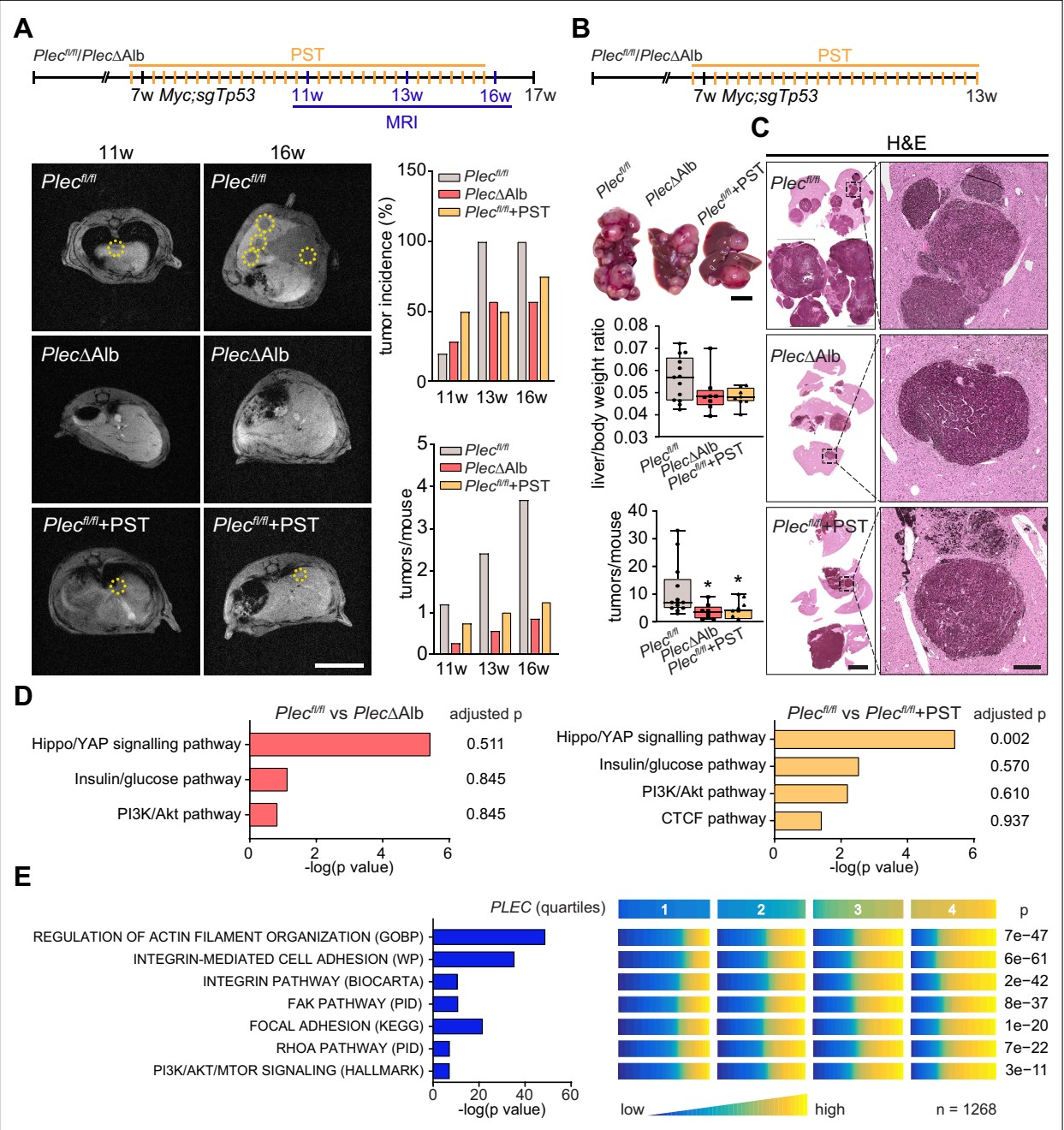

**Figure 7.** Genetic and pharmacological plectin targeting prevents hepatocarcinogenesis through signatures shared by animal models and patients. (**A**) Hepatocellular carcinoma (HCC)-predisposing lesions were introduced by hydrodynamic gene delivery via tail vein injection (HDTVi) of transposon vector encoding *MYC* in conjunction with CRISPR/Cas9 construct targeting *Tp53* (Myc;sgTp53) in *Plec^fl/fl^* and *Plec*ΔAlb cohorts of 7-wk-old male mice. *Plec^fl/fl^* mice were kept either untreated or every second day provided with orogastric gavage of plecstatin (*Plec^fl/fl^*+PST) and the development of HCC was monitored by MRI at 11, 13, and 16 wk, as indicated. Representative MRI images of *Plec^fl/fl^* and *Plec*ΔAlb and *Plec^fl/fl^*+PST tumors acquired at indicated time points. Dashed circles, tumors. Scale bar, 2 cm. Graphs show the average number of tumors (lower graph) and percentual tumor incidence (upper graph) inferred from MRI images. N=5 (*Plec^fl/fl^*), 7 (*Plec*ΔAlb), 4 (*Plec^fl/fl^*+PST). (**B**) Myc;sgTp53 HCC was induced in *Plec^fl/fl^*, *Plec*ΔAlb, and PST-treated *Plec^fl/fl^* (*Plec^fl/fl^*+PST) male mice as in (**A**). Shown are representative images of *Plec^fl/fl^*, *Plec*ΔAlb, and *Plec^fl/fl^*+PST livers from mice with fully developed multifocal HCC sacrificed 6 wk post-induction. Scale bar, 1 cm. Boxplots show tumor burden in the livers expressed as the liver/body weight ratio (upper graph) and number of tumors per mouse (lower graph). The box represents the median, 25th, and 75th percentile with whiskers reaching the last data point; dots, mice; N=12 (*Plec^fl/fl^*), 9 (*Plec*ΔAlb), 10 (*Plec^fl/fl^*+PST). Two-tailed *t*-test; *p<0.05. (**C**) Representative images of H&E-stained *Plec^fl/fl^*, *Plec*ΔAlb, and *Plec^fl/fl^*+PST liver sections. Note darker areas corresponding to HCC lesions. Boxed areas, 12x images. Scale bars, 5 and 1 mm (boxed areas). (**D**) Gene set enrichment analysis of differentially regulated proteins in *Plec^fl/fl^* vs *Plec*ΔAlb and *Plec^fl/fl^* vs *Plec^fl/fl^*+PST livers from the cohort

*Figure 7 continued on next page*

*Figure 7 continued*

shown in (**A**). Prediction of canonical signaling pathways in *Plec^{fl/fl}* vs *PlecΔAlb* (left) and *Plec^{fl/fl}* vs *Plec^{fl/fl}*+PST (right) proteomes. (**E**) Association of plectin-dependent signatures compiled from human HCC-derived cells (see *Figure 3B–E*) and mouse models (see **D**) with *plectin* (*PLEC*) mRNA expression in HCC patients. Right panel shows the levels of selected signatures in patients grouped into quartiles of *PLEC* expression level. N=1268. p-values were generated from an analysis of variance (ANOVA).

The online version of this article includes the following figure supplement(s) for figure 7:

**Figure supplement 1.** Effect of plectin inactivation on hepatocellular carcinoma (HCC) proliferation, apoptosis, and development.

**Figure supplement 2.** Plectin-related expression signatures hepatocellular carcinoma (HCC) from animal models and patients.

**Figure supplement 3.** Plectin signature in hepatocellular carcinoma (HCC) patients.

annotated as 'Integrin pathway', 'FAK pathway', 'PI3K Akt/mTOR signaling', or 'Erk pathway' to positively correlate with elevated plectin expression (*Figure 7E*; *Figure 7—figure supplement 2C*; *Figure 7—figure supplement 3*). Collectively, these data connect plectin with well-characterized pro-oncogenic signaling pathways which were previously identified as prime candidates for therapeutic intervention in cancer (*Cooper and Giancotti, 2019*; *Hoxhaj and Manning, 2020*; *Sun et al., 2016*).

## Discussion

HCC represents a leading cause of cancer-related death, characterized by poor long-term prognosis, high postoperative recurrence, and a high rate of metastasis (*Llovet et al., 2021*; *Singal et al., 2023*). As chemotherapy, surgical resection, radiation, and local ablation are not effective in a large group of patients (*Llovet et al., 2021*; *Ladd et al., 2024*), there is an urgent need to develop effective therapeutic strategies to target HCC. By combining comprehensive analysis of CRISPR/Cas9-engineered HCC cell lines with (phospho)proteomics, mouse modeling as well as human patient data, we identified the plakin family member plectin as a novel HCC marker and druggable target upstream of FAK, MAPK/Erk, and PI3K/Akt signaling. Thus, our data link plectin, a cytolinker implicated in cytoskeletal tension and mechanotransduction with a major oncogenic signaling hub controlling growth and metastasis of HCC.

We began this work by assessing plectin expression in publicly available HCC patient datasets. Our meta-analyses revealed plectin transcript levels to be considerably elevated in HCC irrespective of etiology or gender, whereas previous findings in HCC were inconsistent (*Gundesli et al., 2023*; *Liu et al., 2011*). Notably, we found that patients with higher *PLEC* mRNA levels had significantly shorter recurrence-free survival times than those with lower *PLEC* mRNA levels. Strikingly, *PLEC* expression in publicly available datasets was significantly associated with gene signatures related to "cell survival and proliferation", "angiogenesis", and "hypoxia" (*Figure 7—figure supplement 3*) indicating that the *PLEC* mRNA level was associated with more aggressive cancer traits in HCC patients. In addition, *PLEC* expression levels were associated with TNM staging, underscoring plectin's prognostic value for HCC patient survival. Although HCC transcriptomes appear to differ from other cancers (*Uhlen et al., 2017*), our findings are in line with higher *PLEC* expression in other cancer entities such as oral squamous cell carcinoma (*Flores et al., 2016*; *Yang et al., 2019*), testicular cancer (*Paumard-Hernández et al., 2018*), or pancreatic cancer (*Yin et al., 2021*), and identify plectin as a specific marker for both early and advanced stages of HCC.

We and others have proposed that plectin plays a central role in tumor growth and dissemination (*Perez et al., 2021a*; *Strouhalova et al., 2020*). Here, using liver-specific *PlecΔAlb* knockout mice (*Jirouskova et al., 2018*), we show that plectin ablation in hepatocytes significantly reduced tumor burden in a model of DEN-induced HCC (*Tolba et al., 2015*), which mimics fundamental aspects of human disease (*Lee et al., 2004*). These mice also showed decreased hepatocarcinogenesis in a powerful model of multifocal HCC formation following hydrodynamic delivery of Myc;sgTp53 (*Revia et al., 2022*; *Moon et al., 2019*). In this model, both genetic and PST-mediated pharmacological inactivation of plectin not only reduced the number of HCC tumors formed but ultimately resulted in significantly improved survival of *PlecΔAlb* female mice. Complementing the data from both HCC models, we found that plectin inactivation resulted in the reduced tumorigenic potential of human HCC cells, as evidenced by reduced colony growth under adhesion-independent conditions or subcutaneous xenografts in immunodeficient NSG mice. While several approaches (such as genetic

manipulation [*Buckup et al., 2021*], PST treatment [*Meier et al., 2017*], or blocking peptides [*Pal et al., 2017*] and antibodies [*Perez et al., 2021b*]) decreasing the levels of functional plectin also lead to limited xenograft growth, to our knowledge, this is the first study showing that plectin inactivation prevents tumor progression in well-established preclinical mouse models.

Our previous studies demonstrated that plectin inactivation abrogates physical crosstalk between actin and IF networks (*Prechova et al., 2022*; *Gregor et al., 2014*), leading to the redistribution of internal tension (*Prechova et al., 2022*), and ultimately resulting in defects in cell adhesions (*Gregor et al., 2014*). Indeed, plectin-dependent cytoskeletal disruption and aberrant adhesions have been previously linked to compromised migration and invasion of many non-cancerous (*De Pascalis et al., 2018*; *Gregor et al., 2014*; *Marks et al., 2022*; *Abrahamsberg et al., 2005*; *Zrelski et al., 2024*) as well as cancerous cell types (*Buckup et al., 2021*; *Katada et al., 2012*; *McInroy and Määttä, 2011*; *Sutoh Yoneyama et al., 2014*; *Wenta et al., 2022*), including HCC cells (*Xu et al., 2022*). In support of this concept, we report the collapse of actin and vimentin IF networks in Huh7 and SNU-475 cells with disabled plectin. Cytoskeletal disruption was accompanied by a redistribution of misshapen FAs, which exerted reduced traction forces onto the underlying substrates. As anticipated, aberrant cytoarchitecture resulted in significantly slower motility of HCC cells in both 2D and 3D environments. Consistent with *in vitro* findings, plectin inactivation reduced metastatic outgrowth of HCC cells in the lung. Intriguingly, morphodynamic contour analysis revealed in these cells reduced capacity to form stable protrusions implicated in driving path finding and cellular locomotion (*Bodor et al., 2020*). Collectively, our data suggest that plectin is essential for spatiotemporal cytoskeletal rearrangement, cell shape stabilization, and effective transmission of traction forces, and place plectin-mediated cytoskeletal crosstalk at the center of the processes that control the metastatic cascade.

Plectin-mediated cytoskeletal crosstalk at FAs facilitates their essential features such as dynamics (*Gregor et al., 2014*), adhesion strength (*Bhattacharya et al., 2009*), and mechanotransduction capacity (*Gregor et al., 2014*). Loss of vimentin filament-FA linkage upon plectin deletion in highly migratory dermal fibroblasts was shown to uncouple the activation of FAK from actomyosin-generated tension and attenuate downstream effectors such as Src, Erk1/2, and p38 (*Gregor et al., 2014*). Here, we show that plectin-dependent perturbation of the cytoskeleton-FAs interplay in invasive SNU-475 HCC cells profoundly altered (phospho)proteomic signatures of cytoskeleton- and cell adhesion-annotated proteins, thereby modulating mechanosensitive integrin-associated signaling events. Importantly, our (phospho)proteomic and immunoblot analyses identified attenuated signaling along FAK, MAPK/Erk, and PI3K/Akt axes as a consequence of plectin inactivation in both Huh7 and SNU-475 HCC cells. Plectin's control of cytoskeletal crosstalk and its interplay with pro-oncogenic signaling pathways thus emerges as a critical determinant of the initiation and progression of HCC. It is noteworthy that plectin-dependent effects on PI3K/Akt and FAK/Erk signaling were recently described for prostate cancer (*Katada et al., 2012*; *Wenta et al., 2022*) and head and neck squamous carcinoma cells (*Burch et al., 2013*), indicating that these observations have broader implications beyond liver cancer. Finally, we were able to translate our findings from HCC cell lines and mouse models to HCC patients. By mining data from a large human patient cohort, we found a positive correlation between plectin expression and FA-associated FAK, Erk, and PI3K/Akt pathway gene sets. However, it is conceivable that dysregulated cytoskeletal crosstalk could affect HCC through multiple mechanisms independent by FA-associated signaling. Indeed, we and others (*Jirouskova et al., 2018*; *Xu et al., 2022*) have shown that upon plectin inactivation, liver cells acquire epithelial characteristics that promote increased intercellular cohesion and reduced migration. Further studies will be required to identify and investigate synergistic adhesion-independent effects of plectin inactivation on HCC growth and metastasis.

Current systemic therapies for advanced HCC rely on a combination of multikinase inhibitors (such as sorafenib) or anti-VEGF antibodies/VEGF inhibitors (such as bevacizumab) treatment with immunotherapy (*Cappuyns et al., 2024*). Multikinase inhibitors provide only moderate survival benefit (*Llovet et al., 2018*; *Llovet et al., 2008*) due to primary resistance and the plasticity of signaling networks (*Yau et al., 2008*), and only a subset of patients benefit from the addition of immunotherapy in HCC treatment (*Yau et al., 2019*). Therefore, the most translationally impactful finding of this work is the ability of a small organoruthenium compound PST, a high-affinity plectin ligand, to effectively limit hepatocarcinogenesis in Myc;sgTp53-driven HCC mouse model as well as xenografted human HCCs, leading to the dampening of HCC burden. Using PST, we further report a

marked effect on metastatic HCC outgrowth in the lung along with a reduction of the migratory potential of human HCC cells in 2D and 3D settings. Most notably, our animal models show improvement in local and metastatic survival rates. Similar to other ruthenium-based metallodrugs (*Bakewell et al., 2018*; *Burris et al., 2016*; *Flocke et al., 2016*), PST was well-tolerated by mice and human cells, suggesting good potential for clinical utilization. We and others have previously demonstrated that PST treatment closely mimics phenotypes fostered by ablation of the plectin gene (*Prechova et al., 2022*; *Meier et al., 2017*; *Meier-Menches et al., 2019*; *Wernitznig et al., 2020*). Consistently, PST-mediated inhibition of plectin attenuates FAK, MAPK/Erk, and PI3K/Akt pathways in HCC cells with efficacy comparable to CRISPR/Cas-9-engineered functional (ΔIFBD) or full (KO) knockouts. However, despite high PST target selectivity for plectin (*Meier-Menches et al., 2019*), our data do not rule out pleiotropic effects of PST in the liver and further studies will be required to investigate whether PST mode-of-action in HCC entails molecular mechanisms other than engaging prooncogenic signaling cascades.

## Materials and methods

### Patient tissue samples

Formalin-fixed paraffin-embedded (FFPE) human liver tissue specimens were prepared at the Department of Surgery of the University Hospital Mannheim. The cohort consisted of 21 patients diagnosed with HCC (for details, see *Supplementary file 1*). Tissue collection and analysis were performed in accordance with institutional review board guidelines (reference no. 2012–293 N-MA), and written informed consent was obtained from all included patients.

### Animals

Liver-specific deletion of the plectin (*Plec*) gene was achieved by crossing *Plectin*^flox/flox mice (*Plec*^fl/fl^; *Ackerl et al., 2007*) with *Alb-Cre* transgenic mice (MGI 2176228; The Jackson Laboratory, Bar Harbor, ME) to generate *Plectin*^lox/lox/Alb-Cre^ (*Plec*ΔAlb) mice (*Jirouskova et al., 2018*). Immunodeficient NOD.Cg-Prkdcscid Il2rgtm1Wjl/SzJ (NSG) mice were purchased from the Czech Centre for Phenogenomics (BIOCEV – Institute of Molecular Genetics Academy of Sciences, Prague, Czechia).

Animals were housed under specific pathogen-free conditions with regular access to chow and drinking water and 12 hr light/12 hr dark conditions. All animal studies were performed in accordance with European Directive 86/609/EEC and were approved by the Czech Central Commission for Animal Welfare. Age-matched littermate mice were used in all experiments. The details regarding animal treatments can be found in the sections included in *Supplementary file 3*.

### DEN-induced HCC mouse model

2-week-old *Plec*^fl/fl^ and *Plec*ΔAlb mice received intraperitoneal injection of 25 mg/kg diethylnitrosamine (DEN; Sigma-Aldrich, St. Louis, MO, USA) diluted in PBS. Mice were monitored for tumor formation 30 and 42 wk after the DEN injection by magnetic resonance imaging (MRI) and tumor volumes were calculated from MRI images (for details see the Magnetic Resonance Imaging section included in *Supplementary file 3*). Mice were sacrificed at 44 wk post-injection, livers were dissected, and tumors were measured using a caliper.

### Lung colonization assay

Huh7 and SNU-475 cell lines stably expressing Red Firefly Luciferase reporter and GFP were prepared by lentiviral transfection of LentiGlo pLenti-CMV-RedFluc-IRES-EGFP plasmid (LP-31, Targeting Systems, El Cajon, CA, USA) according to the manufacturer's protocol. Next, 2×10^6 Huh7 or SNU-475 cells suspended in serum-free Dulbecco's modified Eagle medium (DMEM, Sigma-Aldrich) were injected into the tail vein of 5-wk-old NSG mice. The mice were monitored for survival analysis or monitored using bioluminescence imaging for the presence of lung metastasis after 5 wk. Prior to imaging, mice were anesthetized with isoflurane and injected intraperitoneally with D-luciferin potassium salt (Promega, Madison, WI, USA). Ten to fifteen min after injection, luciferase activity was measured using LagoX (Spectral Instruments Imaging, Tuscon, AZ, USA).

## HDTVi-induced HCC mouse model

For hydrodynamic tail vein injections, a mixture of a plasmid mix containing 5 µg/ml of px330 expressing Tp53 sgRNA, 5 µg/ml of pT3-EF1a MYC DNA (92046, Addgene, Watertown, MA, USA), and 0.5 µg/ml pCMV HSB2 sleeping beauty transposase was prepared in a sterile 0.9% sodium chloride (NaCl) solution. 7-wk-old $Plec^{fl/fl}$ and $Plec\Delta Alb$ mice were pre-warmed for 15 min using two infrared lamps (IL 11, Beuer GmbH, Ulm, Germany), placed in a restrainer (TV-RED-150_STD, Braintree Scientific Inc, Braintree, MA, USA) and injected intravenously via the lateral tail vein with a total volume corresponding to 10% of body weight over 5–7 s. All animals were monitored daily, and animal experiments were performed in compliance with all relevant ethical regulations outlined in the animal permit. After mice were sacrificed, livers and lungs were visually inspected, excised, and photographed. Tumor samples were taken to obtain protein, and the remaining liver tissue was incubated in 4% PFA for at least 24 hr for FFPE tissue preparation.

## Statistical analyses

All data mining with the exception of patient analysis, proteomics on mouse tissue samples, and proteomics of SNU-475 cell cultures (see details in corresponding sections), all graphs and statistical tests were performed using GraphPad Prism (GraphPad Software, Inc, La Jolla, CA). In the boxplots, the box margins represent the 25th and 75th percentile with the midline indicating the median. Whiskers reach the last data point. Data comparison of adjacent tumor and non-tumor tissue was performed using a paired *t*-test. Data comparison of individual experimental groups with the control group was performed using a two-tailed *t*-test. Growth curves were analyzed using Two-way ANOVA. Survival curves were analyzed using the Mantel-Cox test. Data distributions were assumed to be normal, but this was not formally tested. Statistical significance was determined at the level of *$p<0.05$, **$p<0.01$, †$p<0.001$. The number of independent experiments (N), number of data points (n), and statistical tests used are specified for individual experiments in the figure legends.

For further details regarding the materials used, please refer to *Supplementary file 3*.

## Acknowledgements

We would like to thank D Tschaharganeh (DKFZ, Heidelberg) for generously providing the px330 (Tp53 sgRNA) and pT3-EF1a MYC plasmids, and B Schuster (IMG CAS, Prague) for pX330 Cas9-Venus plasmid; D Heide and J Hetzer (DKFZ, Heidelberg) for their outstanding technical assistance; B Fabry (FAU Erlangen-Nürnberg), K Volz (DKFZ, Heidelberg), J Prochazka (Czech Centre for Phenogenomics, Vestec), M Maninova, M K Adamcova, M Burocziova, M Capek, and J Valecka (all IMG CAS, Prague) for their expertise. We acknowledge the Light Microscopy Core Facility, IMG CAS, Prague, Czech Republic, for support with advanced microscopy imaging. This work was supported by the Grant Agency of the Czech Republic (GA21-21736S and GA24-10672S); the Institutional Research Project of the Czech Academy of Sciences (RVO 68378050); National Institute for Cancer Research (Programme EXCELES, LX22NPO5102) - Funded by the European Union - Next Generation EU; MEYS CR projects (LM2023050, LM2018126, LQ1604 NPU II, LO1419, and LM2015040); and MEYS CR/ERDF projects (OP RDI CZ.1.05/2.1.00/19.0395 and CZ.1.05/1.1.00/02.0109).

## Additional information

### Funding

| Funder | Grant reference number | Author |
|---|---|---|
| Grantová Agentura České Republiky | GA21-21736S | Martin Gregor |
| Grantová Agentura České Republiky | GA24-10672S | Daniel Rösel |
| Ministerstvo Školství, Mládeže a Tělovýchovy | RVO: 68378050 | Martin Gregor |

| Funder | Grant reference number | Author |
|---|---|---|
| Next Generation EU | LX22NPO5102 | Martin Gregor |
| Ministerstvo Školství, Mládeže a Tělovýchovy | LM2023050 | Martin Gregor |
| Ministerstvo Školství, Mládeže a Tělovýchovy | LM2018126 | Martin Gregor |
| Ministerstvo Školství, Mládeže a Tělovýchovy | LQ1604 | Martin Gregor |
| Ministerstvo Školství, Mládeže a Tělovýchovy | LO1419 | Martin Gregor |
| Ministerstvo Školství, Mládeže a Tělovýchovy | LM2015040 | Martin Gregor |
| Ministerstvo Školství, Mládeže a Tělovýchovy | CZ.1.05/2.1.00/19.0395 | Martin Gregor |
| ERDF | CZ.1.05/1.1.00/02.0109 | Martin Gregor |
| Ministry of Health of the Czech Republic | RVO - 00023001 | Daniel Jirak |
| Next Generation EU | LX22NPO5104 | Daniel Jirak |

The funders had no role in study design, data collection and interpretation, or the decision to submit the work for publication.

## Author contributions

Zuzana Outla, Formal analysis, Investigation, Visualization, Writing – original draft, Writing – review and editing; Gizem Oyman-Eyrilmez, Lenka Sarnova, Piyush Bisht, Petra Novotna, Patricia Bortel, Yasmin Borutzki, Andrea Bileck, Christopher Gerner, Bibiana Kvasnicova, Andrea Galisova, Katerina Sulkova, Investigation; Katerina Korelova, Njainday Jobe, Ondrej Tolde, Eva Sticova, Formal analysis, Investigation; Magdalena Prechova, Formal analysis, Investigation, Visualization; Lukas Frick, Data curation, Formal analysis, Visualization; Jan Kosla, Formal analysis, Investigation, Visualization, Methodology, Writing – review and editing; Mohammad Rahbari, Nuh Rahbari, Emrullah Birgin, Resources; Andreas Bauer, Methodology; Daniel Rösel, Resources, Methodology; Tracy O'Connor, Writing – review and editing; Martin Otahal, Resources, Supervision, Investigation, Methodology; Daniel Jirak, Resources, Formal analysis, Supervision, Methodology; Mathias Heikenwälder, Gerhard Wiche, Resources, Writing – review and editing; Samuel M Meier-Menches, Resources, Data curation, Formal analysis, Supervision, Methodology, Writing – review and editing; Martin Gregor, Conceptualization, Resources, Supervision, Funding acquisition, Visualization, Writing – original draft, Project administration, Writing – review and editing

## Author ORCIDs

Zuzana Outla  https://orcid.org/0000-0002-8216-4724
Daniel Rösel  https://orcid.org/0000-0001-7221-8672
Gerhard Wiche  https://orcid.org/0000-0001-9550-5463
Martin Gregor  https://orcid.org/0000-0001-6841-9527

## Ethics

Formalin-fixed paraffin-embedded (FFPE) human liver tissue specimens were prepared at the Department of Surgery of the University Hospital Mannheim. The cohort consisted of 21 patients diagnosed with HCC (for details, see Supplementary file 1). Tissue collection and analysis were performed in accordance with institutional review board guidelines (reference no. 2012-293N-MA), and written informed consent was obtained from all included patients.

Animals were housed under specific pathogen-free conditions with regular access to chow and drinking water and 12 h light/12 h dark conditions. All animal studies were performed in accordance with European Directive 86/609/EEC and were approved by the Czech Central Commission for Animal Welfare.

Reviewer #1 (Public review): https://doi.org/10.7554/eLife.102205.3.sa1
Reviewer #2 (Public review): https://doi.org/10.7554/eLife.102205.3.sa2
Reviewer #3 (Public review): https://doi.org/10.7554/eLife.102205.3.sa3
Author response https://doi.org/10.7554/eLife.102205.3.sa4

## Additional files

### Supplementary files

Supplementary file 1. Table of patients' clinical data.

Supplementary file 2. List of antibodies used in this study.

Supplementary file 3. Supplemental material.

MDAR checklist

### Data availability

Proteomic data was submitted to the ProteomeXchange Consortium and is available in the PRIDE partner repository (*Perez-Riverol et al., 2025*) with identifiers PXD060086 (*in vitro* profiling), PXD060083 (*in vitro* phosphoproteomics) and PXD060054 (*in vivo* liver).

The following datasets were generated:

| Author(s) | Year | Dataset title | Dataset URL | Database and Identifier |
|---|---|---|---|---|
| Gerner C | 2025 | Plectin-mediated cytoskeletal crosstalk as a target for inhibition of hepatocellular carcinoma growth and metastasis - subcellular *in vitro* profiling | https://www.ebi.ac.uk/pride/archive/projects/PXD060086 | PRIDE, PXD060086 |
| Gerner C | 2025 | Plectin-mediated cytoskeletal crosstalk as a target for inhibition of hepatocellular carcinoma growth and metastasis - *in vitro* phosphoproteomics | https://www.ebi.ac.uk/pride/archive/projects/PXD060083 | PRIDE, PXD060083 |
| Gerner C | 2025 | Plectin-mediated cytoskeletal crosstalk as a target for inhibition of hepatocellular carcinoma growth and metastasis - *in vivo* profiling | https://www.ebi.ac.uk/pride/archive/projects/PXD060054 | PRIDE, PXD060054 |

The following previously published datasets were used:

| Author(s) | Year | Dataset title | Dataset URL | Database and Identifier |
|---|---|---|---|---|
| Wang XW | 2010 | Gene expression data of human hepatocellular carcinoma (HCC) | https://www.ncbi.nlm.nih.gov/geo/query/acc.cgi?acc=GSE14520 | NCBI Gene Expression Omnibus, GSE14520 |
| Yenamandra SP | 2017 | Microarray expression data for tumor and adjacent non-tumor tissues from hepatocellular carcinoma patients | https://www.ncbi.nlm.nih.gov/geo/query/acc.cgi?acc=GSE76427 | NCBI Gene Expression Omnibus, GSE76427 |

*Continued on next page*

*Continued*

| Author(s) | Year | Dataset title | Dataset URL | Database and Identifier |
|---|---|---|---|---|
| Park CK | 2012 | Gene Expression Profiles of both tumor and adjacent non-tumor liver Identify Hepatocellular Carcinoma Patients at High Risk of Recurrence after Curative Hepatectomy | https://www.ncbi.nlm.nih.gov/geo/query/acc.cgi?acc=GSE36376 | NCBI Gene Expression Omnibus, GSE36376 |
| Erickson BJ, Kirk S, Lee Y, Bathe O, Kearns M, Gerdes C, Rieger-Christ K, Lemmerman J | 2016 | The Cancer Genome Atlas Liver Hepatocellular Carcinoma Collection (TCGA-LIHC) | https://doi.org/10.7937/K9/TCIA.2016.IMMQW8UQ | The Cancer Imaging Archive, 10.7937/K9/TCIA.2016.IMMQW8UQ |
| Villanueva A, Llovet JM | 2015 | DNA methylation-based prognosis and epidrivers in hepatocellular carcinoma | https://www.ncbi.nlm.nih.gov/geo/query/acc.cgi?acc=GSE63898 | NCBI Gene Expression Omnibus, GSE63898 |
| Makowska Z | 2016 | Gene expression profiling in paired human hepatocellular carcinoma and liver parenchyma biopsies and normal liver biopsies | https://www.ncbi.nlm.nih.gov/geo/query/acc.cgi?acc=GSE64041 | NCBI Gene Expression Omnibus, GSE64041 |
| Wang XW | 2017 | Gene expression data of human hepatocellular carcinoma (HCC) and Cholangiocarcinoma (CCA) from Thailand Initiative in Genomics and Expression Research for Liver Cancer (TIGER-LC) | https://www.ncbi.nlm.nih.gov/geo/query/acc.cgi?acc=GSE76297 | NCBI Gene Expression Omnibus, GSE76297 |
| Seon-Kyu K | 2010 | Gene expression study in hepatocellular carcinoma | https://www.ncbi.nlm.nih.gov/geo/query/acc.cgi?acc=GSE16757 | NCBI Gene Expression Omnibus, GSE16757 |
| Ivan R | 2010 | Gene expression in nontumoral liver tissue and recurrence-free survival in hepatitis C virus-positive HCC | https://www.ncbi.nlm.nih.gov/geo/query/acc.cgi?acc=GSE17856 | NCBI Gene Expression Omnibus, GSE17856 |
| Hua D | 2015 | Next Generation Sequencing Identification of HBV-MLL4 integration and its molecular basis in Chinese hepatocellular carcinoma | https://www.ncbi.nlm.nih.gov/geo/query/acc.cgi?acc=GSE65485 | NCBI Gene Expression Omnibus, GSE65485 |
| Robert G | 2013 | Expression profiling of HCC | https://www.ncbi.nlm.nih.gov/geo/query/acc.cgi?acc=GSE50579 | NCBI Gene Expression Omnibus, GSE50579 |

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
