## [Editor Report · eLife Assessment]

This **valuable** study investigated the role of PLECTIN, a cytoskeletal crosslinker protein, in hepatocellular carcinoma development and progression. Using a liver-specific Plectin knockout mouse model, the authors showed **solid** evidence that PLECTIN is critical for hepatocarcinogenesis, since inhibition of PLECTIN suppressed tumor formation in multiple models. They also show that PLECTIN is key for HCC invasion and metastasis. They show a correlation between PLECTIN inhibition and attenuated FAK, MAPK/ERK, and PI3K/AKT signaling.

---

## [Referee Report · Reviewer #1 (Public review)]

Summary:

This study investigated the role of PLECTIN, a cytoskeletal crosslinker protein, in liver cancer formation and progression. Using the liver-specific Plectin knockout mouse model, the authors convincingly showed that PLECTIN is critical for hepatocarcinogenesis, as functional inhibition of PLECTIN suppressed tumor formation in several models. They also provided evidence to show that inhibition of PLECTIN inhibited HCC cell invasion and reduced metastatic outgrowth in the lung. Mechanistically, they suggested that PLECTIN inhibition attenuated FAK, MAPK/ERK, and PI3K/AKT signaling.

Strengths:

The authors generated a liver-specific Plectin knockout mouse model. By using DEN and sgP53/MYC models, the authors convincingly demonstrated an oncogenic role of PLECTIN in HCC development. plecstatin-1 (PST), as a plectin inhibitor, showed promising efficacy in inhibiting HCC growth, which provides a basis for potentially treating HCC using PST.

The MIR images for tracking tumor growth in animal models were compelling. The high-quality confocal images and related qualifications convincingly showed the impact of plectin functional inhibition on contractility and adhesions in HCC cells.

Comments on latest version:

My concerns have been largely addressed. The authors did a good job in addressing the questions and clarifying the inconsistent results. I have two comments:

(1) The current data still cannot support the conclusion that plectin inactivation attenuates HCC oncogenic potential through FAK, Erk1/2, and PI3K/Akt axis, unless they can reactivate these signaling to restore the HCC congenic potential in plectin inactivated cells. It might be more appropriate to claim that plectin inactivation suppresses FAK, Erk1/2, and PI3K/Akt oncogenic signaling.

(2) I think it would be beneficial to include the H&E and HNF4α staining from lung tissue of mice inoculated with WT Huh7 cells indicated in the rebuttal letter.

---

## [Referee Report · Reviewer #2 (Public review)]

Summary:

Plectin is a cytolinker that associates with cytoskeletal and intercellular junction proteins and is essential for epithelial integrity and cell migration. Previous reports showed that PLEC regulates tumor growth and metastasis in different cancers. In this manuscript, the authors describe PLEC as a target in initiation and growth of HCC. They show that inhibiting PLEC reduced tumorigenesis in different *in vitro* and *in vivo* HCC models, including in a xenograft model, DEN model, oncogene-induced HCC model and a lung metastasis model. A drug PST had similar effects, a purported Plectin inhibitor, suggesting that PLEC inhibition could be a tumor prevention or treatment strategy. Mechanistically, the authors show that inhibiting PLEC results in a disorganized cytoskeleton, deficiency in cell migration, and changes in cancer-relevant signaling pathways. This study demonstrates the importance of understanding mechanobiology of HCC for the development of new treatment strategies.

Strengths:

(1) This study used a variety of *in vivo* models to explore the role of Plectin in HCC formation and metastasis, which extend beyond the cell line-based studies reported in prior research.

(2) Blocking PLEC disrupts pathways that promote tumors and cell migration, thus preventing tumor progression.

(3) Overall, the anti-cancer phenotype is promising, strengthening the important role of PLEC and related factors in tumor growth and metastasis.

Weaknesses:

(1) There is limited novel mechanistic insights as the effect of inhibiting PLEC on the cytoskeleton, cell migration and related signaling pathways have previously been reported.

(2) The results associated with PST, should be interpretated with caution. Although it is reported as an inhibitor of PLECTIN, and the phenotypes and pathways affected are similar to the knock-out, additional research is needed to support whether it will be safe and specific in treating or preventing HCC.

---

## [Referee Report · Reviewer #3 (Public review)]

Summary:

In this manuscript, Outla Z et al described the analysis of Plectin in HCC pathogenesis. Specifically, it was found that elevated Plectin levels in liver tumors, correlated with poor prognosis for HCC patients. Mechanistically, it showed that Plectin-dependent disruption of cytoskeletal networks leads to the attenuation of oncogenic FAK, MAPK/Erk, and PI3K/AKT signals. Finally, the authors showed that Plectin inhibitor plecstatin-1 (PST) is well-tolerated and capable of overcoming therapy resistance in HCC.

Strengths:

The studies of Plectin are not entirely novel (Pubmed: 36613521). Nevertheless, the current manuscript provides a much more detailed mechanistic study and the results have translational implications. Additional strengths include convincing cell biology data, such as Plectin regulates cytoskeletal networks, and HCC migration/invasion.

Comments on latest version:

The authors have addressed my comments.

---

## [Author Response]

The following is the authors’ response to the original reviews.

**Point-by-point responses to the reviewers' comments:**

All three reviewers found our analysis of focal adhesion-associated oncogenic pathways (Figs 3 and S3) to be inconsistent (Reviewer 1), not convincing/consistent (Reviewer 2, #2), and too variable and not well supported (Reviewer 3, #2). This was probably the basis for the eLife assessment, which stated: “However, the study is incomplete because the downstream molecular activities of PLECTIN that mediate the cancer phenotypes were not fully evaluated.” We agree with the reviewers that the degree of attenuation of the FAK, MAP/Erk, and PI3K/AKT signaling pathways differs depending on the cell line used (Huh7 and SNU-475) and the mode of inactivation (CRISPR/Cas9-generated plectin KO, functional KO (∆IFBD), and organoruthenium-based inhibitor plecstatin-1). However, we do not share the reviewers' skepticism about the unconvincing nature of the data presented.

Several previous studies have shown that plectin inactivation invariably leads to dysregulation of cell adhesions and associated signaling pathways in various cell systems. The molecular mechanisms driving these changes are not fully understood, but the most convincingly supported scenarios are uncoupling of keratin filaments (hemidesmosomes; (Koster et al., 2004)) and vimentin filaments (focal adhesions; (Burgstaller et al., 2010; Gregor et al., 2014)) from adhesion sites in conjunction with altered actomyosin contractility (Osmanagic-Myers et al., 2015; Prechova et al., 2022; Wang et al., 2020). This results in altered morphometry (Wang et al., 2020), dynamics (Gregor et al., 2014), and adhesion strength (Bonakdar et al., 2015) of adhesions. These changes are accompanied by reduced mechanotransduction capacity and attenuation of downstream signaling such as FAK, Src, Erk1/2, and p38 in dermal fibroblasts (Gregor et al., 2014); decrease in pFAK, pSrc, and pPI3K levels in prostate cancer cells (Wenta et al., 2022); increase in pErk and pSrc in keratinocytes (Osmanagic-Myers et al., 2006); decrease in pERK1/2 in HCC cells (Xu et al., 2022) and head and neck squamous carcinoma cells (Katada et al., 2012).

Consistent with these published findings, we show that upon plectin inactivation, the HCC cell line SNU475 exhibits aberrant cytoskeletal organization (vimentin and actin; Figs 4A-D, S4A-F), altered number, topography and morphometry of focal adhesions (Figs 4A, E-G, S4H,I), and ineffective transmission of traction forces (Fig 4H,I). Similar, although not quantified, phenotypes are present in Huh7 with inactivated plectin (data not shown). It is worth noting, that even robust cytoskeletal (e.g. #ventral stress fibers, Fig 4A,D and vimentin architecture, Fig S4A-C) and focal adhesion (%central FA, Fig 4A,E) phenotypes differ significantly between different modes of plectin inactivation and would certainly do so if compared between cell lines. These phenotypes are heterogeneous but not inconsistent. Interestingly, both SNU-475 and Huh7 plectin-inactivated cells show similar functional consequences such as prominent decrease in migration speed (Fig 5B). This suggests that while specific aspects of cytoarchitecture are differentially affected in different cell lines, the functional consequences of plectin inactivation are shared between HCC cell lines.

It is therefore not surprising that the activation status of downstream effectors, resulting from different degrees of cytoskeletal and focal adhesion reconfiguration, is not identical (or even comparable) between cell lines and treatment conditions. Furthermore, we compare highly epithelial (keratin- and almost no vimentin-expressing) Huh7 cells with highly dedifferentiated (low keratin- and high vimentinexpressing) SNU-475 cells, which differ significantly in their cytoskeleton, adhesions, and signaling networks. Alternative approaches to plectin inactivation are not expected to result in the same degree of dysregulation of specific signaling pathways. Effects of adaptation (CRISPR/Cas9-generated KOs and ∆IFBDs), engagement of different binding domains (CRISPR/Cas9-generated ∆IFBDs), and pleiotropic modes of action (plecstatin-1) are expected.

In our study, we provide the reader with an unprecedented complex comparison of adhesion-associated signaling between WT and plectin-inactivated HCC cell lines. First, we compared the proteomes of WT, KO and PST-treated WT SNU-475 cells using MS-based shotgun proteomics and phosphoproteomics (Fig 3A-C). Second, we extensively and quantitatively immunoblotted the major molecular denominators of MS-identified dysregulated pathways (such as “FAK signaling”, “ILK signaling”, and “Integrin signaling”) with the following results. Data (shown in Figs 3D and S3C) are expressed as a percentage of untreated WT, with downregulated values are highlighted in red:

**Author response table 1. sa4table1:** 

FAK expression (to GAPDH)	85 (KO) 84 (/_\1IFBD) 89 (WT+PST) in Huh7
	82 (KO) 71 (/_\1FBD) 79 (WT+PST) in SNU-475
phospho-Tyr397-FAK (to FAK)	110 (KO) 117 (Delta1IFBD) 94 (WT+PST) in Huh7
	104 (KO) 98 (Delta1FBD) 95 (WT+PST) in SNU-475
Akt expression (to GAPDH)	112 (KO) 136 (Delta1IFBD) 94 (WT+PST) in Huh7
	91 (KO) 78 (/_\1FBD) 94 (WT+PST) in SNU-475
phospho-Ser473-Akt (to Akt)	84 (KO) 70 (/_\1IFBD) 86 (WT+PST) in Huh7
	81 (KO) 67 (/_\1FBD) 88 (WT+PST) in SNU-475
Erk1/2 expression (to GAPDH)	119 (KO) 124 (/_\ IFBD) 105 (WT+PST) in Huh7
	128 (KO) 170 (/_\ IFBD) 86 (WT+PST) in SNU-475
phospho-Thr202/Tyr204-Erk (to Erk)	76 (KO) 78 (/_\1IFBD) 90 (WT+PST) in Huh7
	68 (KO) 51 (/_\1FBD) 98 (WT+PST) in SNU-475
ILK expression (to GAPDH)	103 (KO) 102 (Delta1FBD) 87 (WT+PST) in Huh7
	86 (KO) 74 (/_\1FBD) 99 (WT+PST) in SNU-475
PI3K expression (to GAPDH)	108 (KO) 131 (/_\ IFBD) 106 (WT+PST) in Huh7
	86 (KO) 84 (/_\1FBD) 82 (WT+PST) in SNU-475
phospho-p85 (Tyr458)/p55(Tyr199)-PI3K (to PI3K)	67 (KO) 54 (/_\1FBD) 72 (WT+PST) in Huh7
	76 (KO) 88 (/_\IFBD) 80 (WT+PST) in SNU-475

In addition, we show dysregulated expression (mostly downregulation) of focal adhesion constituents ITGβ1 and αv, talin, vinculin, and paxilin which nicely complements fewer and larger focal adhesions in plectin-inactivated HCC cells. In light of these results, we believe that our statement that “Although these alterations were not found systematically in both cell lines and conditions (reflecting thus presumably their distinct differentiation grade and plectin inactivation efficacy), collectively these data confirmed plectin-dependent adhesome remodeling together with attenuation of oncogenic FAK, MAPK/Erk, and PI3K/Akt pathways upon plectin inactivation” (see pages 8-9) is fully supported. Furthermore, in support of the results of MS-based (phospho)proteomic and immunoblot analyses we show strong correlation between plectin expression and the signatures of “Integrin pathway” (R^2^=0.15, p = 2x10^-45^), “FAK pathway” (R^2^=0.11, p = 2x10^-34^), “PI3K Akt/mTOR signaling” (R^2^=0.06, p = 2x10^-20^) or “Erk pathway” (R^2^=0.10, p = 6x10^-30^) in HCC samples from 1268 patients (Fig S7-2C and S7-3).

In conclusion, we show that plectin is required for proper/physiological adhesion-associated signaling pathways in HCC cells. The HCC adhesome and associated pathways are dysregulated upon plectin inactivation and we show context-dependent varying degrees of attenuation of the FAK, MAPK/Erk, and PI3K/Akt pathways. In our view, presenting context-dependent variability in expression/activation of pathway molecular denominators is a trade-off for our intention to address this aspect of plectin inactivation in the complexity of different cell lines, tissues, and modes of inactivation. We prefer rather this complex approach to presenting “more convincing” black-and-white data assessed in a single cell line (Qi et al., 2022) or upon plectin inactivation by a single approach (compare with otherwise excellent studies such as (Xu et al., 2022) or (Buckup et al., 2021)). In fact, unlike the reviewers, we consider this complexity (and the resulting heterogeneity of the data) to be a strength rather than a weakness of our study.

**Reviewer 1:**
(1) The authors suggest that plectin controls oncogenic FAK, MAPK/Erk, and PI3K/Akt signaling in HCC cells, representing the mechanisms by which plectin promotes HCC formation and progression. However, the effect of plectin inactivation on these signaling was inconsistent in Huh7 and SNU-475 cells (Figure 3D), despite similar cell growth inhibition in both cell lines (Figure 2G). For example, pAKT and pERK were only reduced by plectin inhibition in SNU-475 cells but not in Huh7 cells.

We agree with the reviewer that plectin inactivation yields varying degrees of attenuation of the FAK, MAPK/Erk, and PI3K/Akt pathways depending on the cell type (Huh7 vs SNU-475 cells) and mode of plectin inactivation (CRISPR/Cas9-generated plectin KO vs functional KO (∆IFBD) vs organorutheniumbased inhibitor plecstatin-1). This context-dependent heterogeneity in the expression/activation of molecular denominators of signaling pathways reflects different degrees of cytoskeletal (e.g. #ventral stress fibers, Fig 4A,D and vimentin architecture, Fig S4A-C) and focal adhesion (e.g. %central FA, Fig 4A,E) phenotypes under different conditions. We expect, that functional consequences (such as reduced migration and anchorage-independent proliferation) arise from a combination of changes in individual pathways. The sum of often subtle changes will result in comparable effects not only on cell growth, but also on migration or transmission of traction forces. For more detailed comment, please see our response to all Reviewers on the first three pages of this letter.

We believe, that our data show that both pAkt and pErk are attenuated upon plectin inactivation in both Huh7 and SNU-475 cells. The following data (shown in Figs 3D and S3C) are expressed as a percentage of untreated WT, with downregulated values are highlighted in red:

**Author response table 2. sa4table2:** 

phospho-Ser473-Akt (to Akt)	84 (KO) 70 (DeltaIFBD)86 (WT+PST) in Huh7
	81 (KO) 67 (DeltaFBD)88 (WT+PST) in SNU-475
phospho-Thr202/Tyr204-Erk (to Erk)	76 (KO) 78 (DeltaFBD)90 (WT+PST) in Huh7
	68 (KO) 51 (DeltaFBD)98 (WT+PST) in SNU-475

(2) In addition, pFAK was not changed by plectin inhibition in both cells, and the ratio of pFAK/FAK was increased in both cells.

We agree with the reviewer that pFAK/FAK levels are either comparable or slightly higher upon plectin inactivation. However, we believe that our data convincingly show that FAK expression is downregulated in both Huh7 and Snu-475 cells. In our opinion, this results in an overall attenuation of the FAK signaling (see percentage for Normalized pFAKxNormalized FAK), which is expectedly more pronounced in migratory Snu-475 cells. The following data (shown in Figs 3D and S3C) are expressed as a percentage of untreated WT, with downregulated values are highlighted in red:

**Author response table 3. sa4table3:** 

FAK expression (to GAPDH)	85 (KO) 84 (Delta IFBD) 89 (WT+PST) in Huh7
	82 (KO) 71 (Delta IFBD) 79 (WT+PST) in SNU-475
phospho-Tyr397-FAK (to FAK)	110 (KO) 117 (Delta1FBD) 94 (WT+PST) in Huh7
	104 (KO) 98 (Delta1IFBD) 95 (WT+PST) in SNU-475
Normalized pFAKxNormalized FAK	94 (KO) 98 (/_\1 IFBD) 84 (WT+PST) in Huh7
	85 (KO) 70 (DeltaIFBD) 75 (WT+PST) in SNU-475

Given these results, we feel that our statement that “inhibition of plectin attenuates FAK signaling” (pages 8-9) is well supported.

(3) Thus, it is hard to convince me that plectin promotes HCC formation and progression by regulating these signalings.

Previous studies have shown that dysregulation of cell adhesions and attenuation of adhesionassociated FAK, MAPK/Erk, and PI3K/Akt signaling has inhibitory effects on HCC formation and progression. We show that plectin is required for the proper/physiological functioning of adhesionassociated signaling pathways in selected HCC cells. The HCC adhesome and associated pathways are dysregulated upon plectin inactivation and we show context-dependent varying degrees of attenuation of the FAK, MAPK/Erk, and PI3K/Akt pathways. We support these conclusions by providing the reader with proteomic and phosphoproteomic comparisons of adhesion-associated signaling between WT and plectin-inactivated HCC cell lines (Figs 3B,C and S3A,B). We further validate our findings by extensive and quantitative immunoblotting analysis (Figs 3D and S3C). In addition, we show a strong correlation between plectin expression and the signatures of “Integrin pathway” (R^2^=0.15, p = 2x10^-45^), “FAK pathway” (R^2^=0.11, p = 2x10^-34^), “PI3K Akt/mTOR signaling” (R^2^=0.06, p = 2x10^-20^) or “Erk pathway” (R^2^=0.10, p = 6x10^-30^) in HCC samples from 1268 patients (Fig S7E).

Our data and conclusions are fully consistent with previously published studies in HCC cells. For instance, even a mild decrease in FAK levels leads to a significant reduction in colony size (see effects of KD (Gnani et al., 2017) , effects of FAK inhibitor and sorafenib in xenografts (Romito et al., 2021), or effects of inhibitors in soft agars and xenografts (Wang et al., 2016)). Similar effects were observed upon partial Akt inhibition (compare with Akt inhibitors in soft agars (Cuconati et al., 2013; Liu et al., 2020)). Of course, we cannot rule out synergistic plectin-dependent effects mediated via adhesion-independent mechanisms. To identify these mechanisms and to distinguish contribution of various consequences of cytoskeletal dysregulation to phenotypes described in this manuscript would be experimentally challenging and we feel that these studies go beyond the scope of our current study.

As we feel that the adhesion-independent mechanisms were not sufficiently discussed in the original manuscript, we have removed the original sentence “Given the well-established oncogenic activation of these pathways in human cancer(33), our study identifies a new set of potential therapeutic targets.” (page 15) from the Discussion and added the following text: “However, it is conceivable that dysregulated cytoskeletal crosstalk could affect HCC through multiple mechanisms independent from FA-associated signaling. Indeed, we and others (Jirouskova et al., 2018; Xu et al., 2022) have shown that upon plectin inactivation, liver cells acquire epithelial characteristics that promote increased intercellular cohesion and reduced migration. Further studies will be required to identify and investigate synergistic adhesion-independent effects of plectin inactivation on HCC growth and metastasis.” (page 15). See also our response to Reviewer 2, #4 and Reviewer 3, #3 and #4.

(4) The authors claimed that Plectin inactivation inhibits HCC invasion and metastasis using *in vitro* and *in vivo* models. However, the results from *in vivo* models were not as compelling as the *in vitro* data. The lung colonization assay is not an ideal *in vivo* model for studying HCC metastasis and invasion, especially when Plectin inhibition suppresses HCC cell growth and survival. Using an orthotopic model that can metastasize into the lung or spleen could be much more convincing for an essential claim.

We agree with the reviewer that the orthotopic *in vivo* model would be an ideal setting to address HCC metastasis experimentally. There are several published models of HCC extrahepatic metastasis, including an orthotopic model of lung metastasis (Fan et al., 2012; Voisin et al., 2024; You et al., 2016), but to our knowledge, none of these orthotopic models are commonly used in the field. In contrast, the administration of tumor cells via the tail vein of mice is a standard, well-established approach of first choice for modelling lung metastasis in a variety of tumor types (e.g. (Hiratsuka et al., 2011; Jakab et al., 2024; Lu et al., 2020)), including HCC (Jin et al., 2017; Lu et al., 2020; Tao et al., 2015; Zhao et al., 2020).

Furthermore, we do not believe that the use of an orthotopic model would provide a comparable advantage in terms of plectin-mediated effects on metastatic growth compared to tail vein delivery of tumor cells. Importantly, the lung colonization model used in our study allows for the injection of a defined number of HCC cells into the bloodstream, thus eliminating the effect of the primary tumor size on the number of metastasizing cells. To distinguish between effects of plectin inhibition on HCC cell growth/survival and dissemination, we carefully evaluated both the number and volume of lung metastases (Figs 6I and S6C-F). The observed reduction in the number of metastases (Figs 6I and S6D) reflects the initiation/early phase of metastasis formation, which is strongly influenced by the adhesion, migration, and invasion properties of the HCC cells and corresponds well with the phenotypes described after plectin inactivation *in vitro* (Figs 4H,I; 5; 6A-E; S5; and S6A,B). The reduction in the volume of metastases (Figs 6I and S6E) reflects the effects of plectin inhibition on HCC cell growth and metastatic outgrowth and corresponds well with the *in vitro* data shown in Figs 2G,H and S2F,G.

(5) Also, in Figure 6H, histology images of lungs from this experiment need to be shown to understand plectin's effect on metastasis better.

We are grateful to the reviewer for bringing our attention to the lung colonization assay results presented. The description of the experiments in the text of the original manuscript was incorrect. The animals monitored by *in vivo* bioluminescence imaging (shown in Fig 6H) are the same as the mice from which cleared whole lung lobes were analyzed by lattice light sheet fluorescence microscopy (shown in Fig. 6I). The corrected description is now provided in the revised manuscript as follows: “To identify early phase of metastasis formation, we next monitored the HCC cell retention in the lungs using *in vivo* bioluminescence imaging (Fig. 6H). This experimental cohort was expanded for WT-injected mice which were administered PST…” (page 11).

Therefore, lungs from all animals shown in Fig 6H,I were CUBIC-cleared and analyzed by lattice light sheet fluorescence microscopy. As requested by Reviewer 2, Recommendation #1, we provide in the revised manuscript (Fig S6F) “whole slide scan results for all the groups” which could help to understand plectin's effect on metastasis better”. To address the reviewer's concern, we also post-processed cleared and visualized lungs for hematoxylin staining and immunolabeled them for HNF4α. A representative image is shown as a panel A in Author response image 1. Post-processing of CUBIC-cleared and immunolabeled lung lobes resulted in partial tissue destruction and some samples were lost. In addition, as the entire experimental setup was designed for the early phase of metastasis formation, only small Huh7 foci were formed (compared to the larger metastases that developed within 13 weeks after inoculation shown in the panel B). As the IHC for HNF4α provides significantly lower sensitivity compared to the immunofluorescence images provided in the manuscript, we were only able to identify a few HNF4α-positive foci. Overall, we consider our immunofluorescence images to be qualitatively and quantitatively superior to IHC sections. However, if the reviewer or the editor considers it beneficial, we are prepared to show our current data as a part of the manuscript.

**Author response image 1. sa4fig1:** (**A**) HNF4α staining of lung tissue after CUBIC clearing from mice inoculated with WT Huh7 from the timepoint of BLI, when the positive signal in chest area has been detected. This timepoint was then selected for the comparison of initial stages of lung colonization. (**B**) H&E and HNF4α staining from lung tissue of mice inoculated with WT Huh7 cells from the survival experiment. Scale bars, 50 µm.

(6) Figure 6G, it is unclear how many mice were used for this experiment. Did these mice die due to the tumor burdens in the lungs?

The number of animals is given in the legend to Fig 6G (page 34; N = 14 (WT), 13 (KO)). Large Huh7 metastases were identified in the lungs of animals that could be analyzed post-mortem by IHC (see panel B in the figure above). No large metastases were found in other organs examined, such as the liver, kidney and brain. It is therefore highly likely that these mice died as a result of the tumor burden in the lungs. A similar conclusion was drawn from the results of the lung colonization model in the previous studies (Jin et al., 2017; Zhao et al., 2020).

(7) The whole paper used inhibition strategies to understand the function of plectin. However, the expression of plectin in Huh7 cells is low (Figure 1D). It might be more appropriate to overexpress plectin in this cell line or others with low plectin expression to examine the effect on HCC cell growth and migration.

For this study, we selected two model HCC cell lines – Huh7 and SNU-475. Our intention was to investigate the role of plectin in “well-differentiated” (Huh7) and “poorly differentiated” (SNU-475) HCC cells, including thus early and advanced stages of HCC development (as categorized before Boyault et al., 2007; Yuzugullu et al., 2009a); see also our description and rationale on page 6. As anticipated, less migratory “epithelial-like” Huh7 cells are characterized by relatively high E-cadherin, low vimentin, and low plectin expression levels (Fig 1D). In contrast, migratory “mesenchymal-like” SNU-475 cells are characterized by relatively low E-cadherin, high vimentin, and high plectin expression levels (Fig 1D). Therefore, the majority of analyses were performed in both relatively low plectin-expressing Huh7 and high plectin-expressing SNU-475 cells. It is noteworthy, that inactivation of plectin had similar (although less pronounced) inhibitory effects on growth and migration in both Huh7 and SNU-475 cells.

We agree with the reviewer that “It might be more appropriate to overexpress plectin in this cell line or others with low plectin expression to examine the effect on HCC cell growth and migration”. In fact, we have received similar suggestions since we started publishing our studies on plectin. There are two reasons, which preclude the successful overexpression experiments. First, there are about 14 known isoforms of plectin (Prechova et al., 2023). Although, previous studies have analyzed the phenotypic rescue potential of some plectin isoforms using transient transfection (e.g. (Burgstaller et al., 2010; Osmanagic-Myers et al., 2015; Prechova et al., 2022)), the isoform variability precludes rescue/overexpression experiments if the causative isoform is not known. Second, plectin is a giant cytoskeletal crosslinker protein of more than 4,500 amino acids with binding sites for intermediate filaments, F-actin, and microtubules. Overexpression of the approximately 500 kDa-large crosslinker invariably leads to the collapse of cytoskeletal networks in every cell type we have tested so far. See also our response to Reviewer 3, #2.

**Reviewer 2:**
(1) The annotation of mouse numbers is confusing. In Figures 2A B D E F, it should be the same experiment, but the N numbers in A are 6 and 5. In E and F they are 8 and 3. Similarly, in Figure 2H, in the tumor size curve, the N values are 4,4,5,6. In the table, N values are 8,8,10,11 (the authors showed 8,7,8,7 tumors that formed in the picture).

We are grateful to the reviewer for bringing our attention to the inconsistency the number of animals in DEN-induced hepatocarcinogenesis. Results from two independent cohorts are presented in the manuscript. The first cohort was used for MRI screening (Fig 2A-C) and at the second screening timepoint of 44 weeks, approximately 75% of animals died during anesthesia. Therefore, the second cohort of *Plec*ΔAlb and *Plec*^*fl/fl*^ mice was used for macroscopic confirmation and histology (Figs 2D-F and S2A). We agree with the reviewer that the original presentation of the data may be misleading; therefore, we have rephrased the sentence describing macroscopic confirmation and histology (Figs 2D-F and S2A) as follows: “Decreased tumor burden in the second cohort of *Plec*ΔAlb mice was confirmed macroscopically…” (page 7).

For the experiments shown in Fig 2H, mice were injected in both hind flanks. We have added this information to the figure legend along with the correct number of tumors.

(2) In Figure 3D and Figure S3C, the changes in most of the proteins/phosphorylation sites are not convincing/consistent. These data are not essential for the conclusion of the paper and WB is semi-quantitative. Maybe including more plots of the proteins from proteomic data could strengthen their detailed conclusions about the link between Plectin and the FAK, MAPK/Erk, PI3K/Akt pathways as shown in 3E.

We agree with the reviewer that plectin inactivation yields varying degrees of attenuation of the FAK, MAPK/Erk, and PI3K/Akt pathways depending on the cell type (Huh7 vs SNU-475 cells) and mode of plectin inactivation (CRISPR/Cas9-generated plectin KO vs functional KO (∆IFBD) vs organorutheniumbased inhibitor plecstatin-1). This context-dependent heterogeneity in the expression/activation of pathway molecular denominators reflects different degrees of cytoskeletal (e.g. #ventral stress fibers, Fig 4A,D and vimentin architecture, Fig S4A-C) and focal adhesion (e.g. %central FA, Fig 4A,E) phenotypes under different conditions. See also the detailed response to all reviewers (on the first three pages of this letter) and the responses to Reviewer 1, #1 and #2, Reviewer 3, #4.

Our immunoblot analysis is based on NIR fluorescent secondary antibodies which were detected and quantified using an Odyssey imaging system (LI-COR Biosciences). This approach allows a wider linear detection range than chemiluminescence without a signal loss and is considered to provide quantitative immunoblot detection (Mathews et al., 2009; Pillai-Kastoori et al., 2020) (see also manufacturer's website: https://www.licor.com/bio/applications/quantitative-western-blots/).

Following the reviewer's recommendation, we have carefully reviewed our proteomic and phosphoproteomic data. There are no further MS-based data (other than those already presented in the manuscript) to support the association of plectin with the FAK, MAPK/Erk, PI3K/Akt pathways.

(3) Figure S7A and B, The pictures do not show any tumor, which is different from Figure 7A and B (and from the quantification in S7A lower right). Is it just because male mice were used in Figure 7 and female mice were used in Figure S7? Is there literature supporting the sex difference for the Myc-sgP53 model?

As indicated in the Figure legends and in the corresponding text in the Results section (page 12), the Fig 7A,B shows Myc;sgTp53-driven hepatocarcinogenesis in male mice, whereas Fig S7C,D shows results from the female cohort. In general, the HDTVi-induced HCC onset and progression differs considerably between individual experiments, and it is therefore crucial to compare data within an experimental cohort (as we have done for *Plec*ΔAlb and *Plec^fl/fl^* mice). Nevertheless, we cannot exclude the influence of sexual dimorphism on the results presented. The existence of sexual dimorphism in liver cancer is supported by a substantial body of evidence derived from various studies (e.g. (Bigsby and CaperellGrant, 2011; Bray et al., 2024)). To date, no reports have specifically addressed sexual dimorphism in Myc;sgTp53 HDTVI-induced liver cancer. This is likely due to the fact that the vast majority of studies using this model have only presented data for one sex. However, a study using an HDTVI-administered combination of c-MET and mutated beta-catenin oncogenes to induce HCC in mice observed elevated levels of alpha-fetoprotein (AFP) in males when compared to females (Bernal et al., 2024). The study suggests that estrogen may have a protective effect in female mice, as ovariectomized females had AFP levels comparable to those observed in males. Our data suggest that female hormones may have a similar effect in the Myc;sgTp53 HDTVI-induced liver cancer model.

(4) Figure 2F, S2A, *Plec*ΔAlb mice more frequently formed larger tumors, as reflected by overall tumor size increase. The interpretation of the authors is "possibly implying reduced migration or increased cohesion of plectin-depleted cells". It is quite arbitrary to make this suggestion in the absence of substantial data or literature to support this theory.

We agree with the reviewer that our statement “Notably, *Plec*ΔAlb mice more frequently formed larger tumors, as reflected by overall tumor size increase (Fig. 2F; Figure 2—figure supplement 1A), possibly implying reduced migration or increased cohesion of plectin-depleted cells(25).” (page 7) is rather speculative. As we did not further address the formation of larger tumors in *Plec*ΔAlb mice further in the current study, we wanted to provide the readers with some, even speculative, hypotheses. In support of our hypothesis, we cite our own publication (#26; Jirouskova et al., J Hepatol., 2018), where we show that plectin inactivation in *Plec*ΔAlb livers results in upregulation of the epithelial marker E-cadherin. Previous studies have shown that similar increase in E-cadherin expression levels reflects mesenchymalto-epithelial transition (e.g. (Adhikary et al., 2014; Auersperg et al., 1999; Wendt et al., 2011)) and is often associated with reduced cancer cell migration/invasion. This is consistent with our finding that “migrating plectin-disabled SNU-475 cells exhibited more cohesive, epithelial-like features while progressing collectively. By contrast, WT SNU-475 leader cells were more polarized and found to migrate into scratch areas more frequently than their plectin-deficient counterparts (Figure 5—figure supplement 1B). Consistent with this observation, individually seeded SNU-475 cells less frequently assumed a polarized, mesenchymal-like shape upon plectin inactivation in both 2D and 3D environments (Fig. 5C). Moreover, plectin-inactivated SNU-475 cells exhibited a decrease in N-cadherin and vimentin levels when compared to WT counterparts (Figure 5—figure supplement 1C).” (page 10).

In conclusion, we have shown that plectin-deficient hepatocytes express higher levels of E-cadherin and hepatocyte-derived SNU-475 cells express less N-cadherin and vimentin. In addition, we show that SNU475 cells exhibited more cohesive, epithelial-like features in scratch-wound experiments. To address the reviewer's concern and to further support our statement about the increased cohesiveness of plectindeficient HCC cells we have included the citation of the recent study #27 (Xu et al., 2022). Using the MHCC97H and MHCC97L HCC cell lines, this study shows that plectin downregulation “inhibits HCC cell migration and epithelial mesenchymal transformation”, which is fully consistent with our hypothesis. To mitigate the impression of an unsubstantiated statement, we also discuss adhesion-independent plectin-mediated mechanisms in the revised Discussion section as follows: “However, it is conceivable that dysregulated cytoskeletal crosstalk could affect HCC through multiple mechanisms independent from FA-associated signaling. Indeed, we and others (Jirouskova et al., 2018; Xu et al., 2022) have shown that upon plectin inactivation, liver cells acquire epithelial characteristics that promote increased intercellular cohesion and reduced migration. Further studies will be required to identify and investigate synergistic adhesion-independent effects of plectin inactivation on HCC growth and metastasis.” (page 15).

(5) Mutation or KO PLEC has been shown to cause severe diseases in humans and mice, including skin blistering, muscular dystrophy, and progressive familial intrahepatic cholestasis. Please elaborate on the potential side effects of targeting Plectin to treat HCC.

Indeed, mutation or ablation of plectin has been implicated in many diseases (collectively known as plectinopathies). These multisystem disorders include an autosomal dominant form of epidermolysis bullosa simplex (EBS), limb-girdle muscular dystrophy, aplasia cutis congenita, and an autosomal recessive form of EBS that may be associated with muscular dystrophy, pyloric atresia, and/or congenital myasthenic syndrome. Several mutations have also been associated with cardiomyopathy and malignant arrhythmias. Progressive familial intrahepatic cholestasis has also been reported. In genetic mouse models, loss of plectin leads to skin fragility, extensive intestinal lesions, instability of the biliary epithelium, and progressive muscle wasting (for more details see (Vahidnezhad et al., 2022)).

It is therefore important to evaluate potential side effects, and plectin inactivation therefore presents challenges comparable to other anti-HCC targets. For instance, Sorafenib, the most widely used chemotherapy in recent decades, targets numerous serine/threonine and tyrosine kinases (RAF1, BRAF, VEGFR 1, 2, 3, PDGFR, KIT, FLT3, FGFR1, and RET) that are critical for proper non-pathological functions (Strumberg et al., 2007; Wilhelm et al., 2006; Wilhelm et al., 2004). The combinatorial therapy of atezolizumab and bevacizumab targets also PD-L1 in conjunction with VEGF, which plays an essential role in bone formation (Gerber et al., 1999), hematopoiesis (Ferrara et al., 1996), or wound healing (Chintalgattu et al., 2003). To allow readers to read a comprehensive account of the pathological consequences of plectin inactivation, we included two additional citations (Prechova et al., 2023; Vahidnezhad et al., 2022) and rephrased Introduction section as follows: “…multiple reports have linked plectin with tumor malignancy(12) and other pathologies (Prechova et al., 2023; Vahidnezhad et al., 2022), mechanistic insights…” (page 4-5).

**Reviewer 3:**
(1) The rationale for using Huh7 cells in the manuscript is not well explained as it has the lowest Plectin expression levels.

For this study, we selected two model HCC cell lines - Huh7 and SNU-475. Our intention was to address the role of plectin in “well-differentiated” (Huh7) and “poorly differentiated” (SNU-475) HCC cells, thus including early and advanced stages of HCC development (as categorized before Boyault et al., 2007; Yuzugullu et al., 2009b see also our description and reasoning on page 6). The Huh7 cell line is also a well-established and widely used model suitable for both *in vitro* and *in vivo* settings (e.g. Du et al., 2024; Fu et al., 2018; Si et al., 2023; Zheng et al., 2018).

As anticipated, less migratory “epithelial-like” Huh7 cells are characterized by relatively high E-cadherin, low vimentin, and low plectin expression levels (Fig 1D). In contrast, migratory “mesenchymal-like” SNU475 cells are characterized by relatively low E-cadherin, high vimentin, and high plectin expression levels (Fig 1D). Therefore, the majority of analyses were performed in both relatively low plectin-expressing Huh7 and high plectin-expressing SNU-475 cells. It is noteworthy, that inactivation of plectin had similar (although less pronounced) inhibitory effects on the phenotypes in both Huh7 and SNU-475 cells. We believe that these findings highlight the importance of plectin in HCC growth and metastasis, as plectin inactivation has inhibitory effects on both early (low plectin) and advanced (high plectin) stages of HCC.

(2) The KO cell experiments should be supplemented with overexpression experiments.

We agree with the reviewer that it would be helpful to complement our plectin inactivation experiments by overexpressing plectin in the HCC cell lines used in this study. In fact, we have received similar suggestions since we started to publish our studies on plectin. There are two reasons, which preclude the successful overexpression experiments. First, there is about 14 known isoforms of plectin (Prechova et al., 2023). Although previous studies have analyzed the phenotypic rescue potential of some plectin isoforms using transient transfection (e.g. (Burgstaller et al., 2010; Osmanagic-Myers et al., 2015; Prechova et al., 2022)), the isoform variability precludes rescue/overexpression experiments if the causative isoform is not known. Second, plectin is a giant cytoskeletal crosslinker protein of more than 4,500 amino acids with binding sites for intermediate filaments, F-actin, and microtubules. Overexpression of the approximately 500 kDa-large crosslinker invariably leads to the collapse of cytoskeletal networks in every cell type we have tested so far. See also our response to Reviewer 1, #7.

(3) There is significant concern that while ablation of Ple led to reduced tumor number, these mice had larger tumors. The data indicate that Plectin may have distinct roles in HCC initiation versus progression. The data are not well explained and do not fully support that Plectin promotes hepatocarcinogenesis.

In the DEN-induced HCC model MRI screening revealed fewer tumors and also tumor volume was reduced at 32 and 44 weeks post-induction (Fig 2A-C). Larger tumors formed in *Plec*ΔAlb compared to *Plec^fl/fl^* livers (Figs 2F and S2A) refer only to a subset of macroscopic tumors visually identified at necropsy. Larger *Plec*ΔAlb tumors were not observed in the Myc;sgTp53 HDTVI-induced HCC model (data not shown). In contrast, plectin deficiency reduced the size of xenografts formed in NSG mice (Fig 2H), and agar colonies grown from Huh7 and SNU-475 cells with inactivated plectin were also smaller (Fig S2F). In all *in vivo* and *in vitro* approaches presented in the manuscript, plectin inactivation reduced the number of colonies/xenografts/tumors. As hepatocarcinogenesis is a multistep process including initiation, promotion, and progression (Pitot, 2001), we feel confident in concluding that plectin inactivation inhibits hepatocarcinogenesis and we consider this conclusion to be fully supported by the data presented in the manuscript.

However, we agree with the reviewer that larger macroscopic *Plec*ΔAlb tumors in the DEN-induced HCC model are intriguing. As we do not see similar effects (or even trends) in other approaches used in this study, we cannot exclude the contribution of plectin-deficient environment in *Plec*ΔAlb livers during longterm (44 weeks) tumor formation and growth. In our previous study (Jirouskova et al., 2018), we showed that plectin deficiency in *Plec*ΔAlb livers leads to biliary tree malformations, collapse of bile ducts and ductules, and mild ductular reaction. We could speculate that *Plec*ΔAlb livers suffer from continuous bile leakage into the parenchyma, which would exacerbate all models of long-term pathology.

As we did not further address the formation of larger tumors in *Plec*ΔAlb mice further in the current study, we offered the reader the hypothesis that large tumors could “…possibly implying reduced migration or increased cohesion of plectin-depleted cells25.” In support of our hypothesis, we cite our own publication (#26; Jirouskova et al., J Hepatol., 2018), where we show that plectin inactivation in *Plec*ΔAlb livers results in upregulation of the epithelial marker E-cadherin. Previous studies have shown that similar increase in E-cadherin expression levels reflects mesenchymal-to-epithelial transition (e.g. (Adhikary et al., 2014; Auersperg et al., 1999; Wendt et al., 2011)) and is often associated with reduced cancer cell migration/invasion. This is consistent with our finding that “migrating plectin-disabled SNU475 cells exhibited more cohesive, epithelial-like features while progressing collectively. By contrast, WT SNU-475 leader cells were more polarized and found to migrate into scratch areas more frequently than their plectin-deficient counterparts (Figure 5—figure supplement 1B). Consistent with this observation, individually seeded SNU-475 cells less frequently assumed a polarized, mesenchymal-like shape upon plectin inactivation in both 2D and 3D environments (Fig. 5C). Moreover, plectin-inactivated SNU-475 cells exhibited a decrease in N-cadherin and vimentin levels when compared to WT counterparts (Figure 5—figure supplement 1C).” (page 10).

In conclusion, we have shown that plectin-deficient hepatocytes express higher levels of E-cadherin and hepatocyte-derived SNU-475 cells less N-cadherin and vimentin. In addition, we show that SNU-475 cells exhibited more cohesive, epithelial-like features in scratch-wound experiments. To address the reviewer's concern and to further support our claim of increased cohesiveness of plectin-deficient HCC cells we included the citation of the recent study(27). Using the MHCC97H and MHCC97L HCC cell lines, this study shows that plectin downregulation “inhibits HCC cell migration and epithelial mesenchymal transformation” and is therefore fully consistent with our hypothesis. To mitigate the impression of an unsubstantiated statement, we also discuss adhesion-independent plectin-mediated mechanisms in the revised Discussion section as follows: “However, it is conceivable that dysregulated cytoskeletal crosstalk could affect HCC through multiple mechanisms independent from FA-associated signaling. Indeed, we and others (Jirouskova et al., 2018; Xu et al., 2022) have shown that upon plectin inactivation, liver cells acquire epithelial characteristics that promote increased intercellular cohesion and reduced migration. Further studies will be required to identify and investigate synergistic adhesionindependent effects of plectin inactivation on HCC growth and metastasis.” (page 15).

(4) Figure 3 showed that Plectin does not regulate p-FAK/FAK expression. Therefore, the statement that Plectin regulates the FAK pathway is not valid. Furthermore, there are too many variables in turns of p-AKT and p-ERK expression, making the conclusion not well supported.

We agree with the reviewer that pFAK/FAK levels are either comparable or slightly higher upon plectin inactivation. However, we believe that our data convincingly show that FAK expression is downregulated in both Huh7 and Snu-475 cells. In our opinion, this results in an overall attenuation of the FAK signaling (see percentage for Normalized pFAKxNormalized FAK), which is expectedly more pronounced in migratory Snu-475 cells. The following data (shown in Figs 3D and S3C) are expressed as a percentage of untreated WT, with downregulated values highlighted in red:

**Author response table 4. sa4table4:** 

FAK expression (to GAPDH)	85 (KO) 84 (DeltaIFBD)89 (WT+PST) in Huh7
phospho-Tyr397-FAK (to FAK)	82 (KO) 71 (DeltaIFBD)79 (WT+PST) in SNU-475
Normalized pFAKxNormalized FAK	110 (KO) 117 (DeltaIFBD)94 (WT+PST) in Huh7
	104 (KO) 98 (DeltaIFBD)95 (WT+PST) in SNU-475
	94 (KO) 98 (DeltaIFBD)84 (WT+PST) in Huh7
	85 (KO) 70 (DeltaIFBD)75 (WT+PST) in SNU-475

Given these results, we believe that our statement that “inhibition of plectin attenuates FAK signaling” (pages 8-9) is well supported.

We believe, that our data show that both pAkt and pErk are attenuated upon plectin inactivation in both Huh7 and SNU-475 cells. The following data (presented in Figs 3D and S3C) are shown as a percentage of untreated WT, with downregulated values highlighted in red:

**Author response table 5. sa4table5:** 

phospho-Ser473-Akt (to Akt)	84 (KO) 70 (Delta IFBD) 86 (WT+PST) in Huh7
	81 (KO) 67 (Delta IFBD) 88 (WT+PST) in SNU-475
phospho-Thr202/Tyr204-Erk (to Erk)	76 (KO) 78 (Delta IFBD) 90 (WT+PST) in Huh7
	68 (KO) 51 (Delta IFBD) 98 (WT+PST) in SNU-475

We agree with the reviewer that plectin inactivation yields varying degrees of attenuation of the FAK, MAPK/Erk, and PI3K/Akt pathways depending on the cell type (Huh7 vs SNU-475 cells) and mode of plectin inactivation (CRISPR/Cas9-generated plectin KO vs functional KO (∆IFBD) vs organorutheniumbased inhibitor plecstatin-1). This context-dependent heterogeneity in the expression/activation of pathway molecular denominators reflects different degrees of cytoskeletal (e.g. #ventral stress fibers, Fig 4A,D and vimentin architecture, Fig S4A-C) and focal adhesion (e.g. %central FA, Fig 4A,E) phenotypes under different conditions. See also the detailed response to all Reviewers (on the first three pages of this letter) and the responses to Reviewer 1, #1 and #2 and Reviewer 2, #4.

(5) The studies of plecstatin-1 in HCC should be expanded to a panel of human HCC cells with various Plectin expression levels in turns of cell growth and cell migration. The IC50 values should be determined and correlate with Plectin expression.

Following the reviewer's suggestion, we have included graphs showing IC50 values for Huh7 (low plectin) and SNU-475 (high plectin) cells as Fig S2E. As expected, the IC50 values are higher for SNU-475 cells. Corresponding parts of the Figure legends have been changed. We refer to new data in the Results section as follows: “If not stated otherwise, we applied PST in the final concentration of 8 µM, which corresponds to the 25% of IC50 for Huh7 cells (Figure 2—figure supplement 1E).” (page 7). We also provide details of the IC50 determination in the revised Supplement Materials and methods section (pages 5-6).

(6) One of the major issues is the mechanistic studies focusing on Plectin regulating HCC migration/metastasis, whereas the *in vivo* mouse studies focus on HCC formation (Figures 3 and 7). These are distinct processes and should not be mixed.

In our study, we investigated the role of plectin in the development and dissemination of HCC. Using DEN- and Myc;sgTp53 HDTVI-induced HCC models (Figs 2A-F, S2A, 7A-C, and S7A-D), we show the effects of plectin inactivation on HCC formation *in vivo*. These studies are complemented by xenografts (Figs 2H and S2G) and *in vitro* colony formation assay (Figs 2G and S2F). Using an *in vivo* lung colonization assay (Figs 6G-I and S6C-F), we show the effects of plectin inactivation on the metastatic potential of HCC cells. In complementary *in vitro* studies, we show how plectin deficiency affects migration (Figs 5 and S5) and invasion (Figs 6A-E and S6A,B).

Our mechanistic studies show that plectin inactivation leads to dysregulation of cytoskeletal networks, adhesions, and adhesion-associated signaling. We believe that we have provided substantial experimental data suggesting that the proposed mechanisms play a role in plectin-mediated inhibition of both HCC development and dissemination. Of course, we cannot rule out additional, adhesionindependent mechanisms for HCC formation. To clarify this, we have revised the Discussion section as follows: “However, it is conceivable that dysregulated cytoskeletal crosstalk could affect HCC through multiple mechanisms independent from FA-associated signaling. Indeed, we and others Jirouskova et al., 2018; Xu et al., 2022 have shown that upon plectin inactivation, liver cells acquire epithelial characteristics that promote increased intercellular cohesion and reduced migration. Further studies will be required to identify and investigate synergistic adhesion-independent effects of plectin inactivation on HCC growth and metastasis.” (page 15).

(7) Figure 7B showed that Ple KO mice were treated with PST, but the data are not presented in the manuscript. Tumor cell proliferation and apoptosis rates should be analyzed as well.

We do not show any effects of PST in *Plec*ΔAlb mice. As stated in the Fig 7B legend: “Myc;sgTp53 HCC was induced in *Plec^fl/fl^*, *Plec*ΔAlb, and PST-treated *Plec^fl/fl^* (*Plec^fl/fl^*+PST) male mice as in (A). Shown are representative images of *Plec^fl/fl^*, *Plec*ΔAlb, and *Plec^fl/fl^*+PST livers from mice with fully developed multifocal HCC sacrificed 6 weeks post-induction.”.

Following the reviewer's recommendation, we include the analysis of proliferation and apoptosis rates as revised Fig S7A,B. Please note, that no differences in apoptosis and proliferation rates were found between experimental conditions. Due to additional data, the original Fig S7 – 1 has been split into revised Fig S7 – 1 and Fig S7 – 2.

(8) The status of FAK, AKT, and ERK pathway activation was not analyzed in mouse liver samples. In Figure 7D, most of the adjusted p-values are not significant.

We are aware that the majority of FDR corrected p-values shown in the Fig 7D are not significant. In fact, we deliberated with our colleagues from the laboratory of Prof. Samuel Meier-Menches (Department of Analytical Chemistry, University of Vienna), who conducted all the proteomic studies presented in this manuscript, on whether to present such "weak" data. Following a lengthy discussion, a decision was taken to include them despite the anticipation of criticism from the reviewers. The rationale for including these data is that, despite the lack of statistical significance, the findings are consistent with those of MS/immunoblot analyses of HCC cells (Figs 3 and S3) and patient data (Figs 7E, S7-2). The lack of statistical significance observed in the presented data is a consequence of the limited number of animals included in the *Plec^fl/fl^*, *Plec*ΔAlb, and PST-treated *Plec^fl/fl^* cohorts, which has resulted in a high degree of variability in the MS results. We agree with the reviewer that the inclusion of immunoblot analysis would provide further support for our conclusions. However, we do not have any remaining liver tissue that could be analyzed.

(9) There is no evidence to support that PST is capable of overcoming therapy resistance in HCC. For example, no comparison with the current standard care was provided in the preclinical studies.

We are grateful to the reviewer for bringing our attention to the incorrect statement in the Abstract: “…we show that plectin inhibitor plecstatin-1 (PST) is well-tolerated and capable of overcoming therapy resistance in HCC”. To address the reviewer's concern, we rephrased the Abstract as follows: “…we show that plectin inhibitor plecstatin-1 (PST) is well-tolerated and potently inhibits HCC progression”.

**Recommendations for the authors:**

**Reviewer 2 (Recommendations for the authors):**
(1) In Figures 6I and S6C, it would be better to show the whole slide scan result for all the groups.

Following the reviewer's recommendation, we include the whole slide scan result for all the groups as revised Fig S6F.

(2) In Figures S7C and D, what do the highlighted/colored dots represent? They are not mentioned in the figure legend or the results.

Following the reviewer's recommendation, we include the explanation in the revised Figure legends (page 30).

(3) In Figure 2H, the experiment schedule showed "6w Huh7 t.v.i.", but should it be subcutaneous injection?

We are grateful to the reviewer for bringing our attention to the incorrect description of the experiment. The schematics was corrected. The schematic has been corrected. We have also noticed an error in the table summarizing the number of tumors formed (N) and have corrected the values for the WT+PST and KO conditions.

(4) Supplemental Materials and Methods, Xenograft tumorigenesis, Error: 2.5×106 Huh7 cells in 250 ml PBS mice were administered subcutaneously in the left and right hind flanks. It probably should be "250ul".

We are grateful to the reviewer for bringing our attention to the incorrect description of the experiment. The corresponding part of the Materials and Methods section has been corrected (page 2).

(5) In Figure legend Supplementary Figure 6 C,D,E : "Representative magnified images from lung lobes with GFP-positive WT, KO, and WT+PST SNU-475 nodules". There is no picture for the WT+PST SNU-475 group.

We are grateful to the reviewer for bringing our attention to the incorrect description of the experiment. The corresponding part of the Figure legend (“*WT+PST SNU-475*”) has been deleted (page 27).

(6) In the Figure legend for Figure 6H, "Representative BLI images of WT, KO, and PST-treated WT (WT+PST) SNU-475 cells-bearing mice are shown". Should it be Huh7, not SNU-475?

We are grateful to the reviewer for bringing our attention to the incorrect description of the experiment. The description of the cell line has been corrected (page 34).

(7) The statement that current therapies rely on multikinase inhibitors is no longer correct.

We are grateful to the reviewer for bringing our attention to the incorrect statement. To address the reviewer's concern, we rephrased the original part of Discussion section: “Current therapies for HCC rely on multikinase inhibitors (such as sorafenib) that provide only moderate survival benefit(60,61) due to primary resistance and the plasticity of signaling networks(62)” as follows: “Current systemic therapies for advanced HCC rely on a combination of multikinase inhibitor (such as sorafenib) or anti-VEGF /VEGF inhibitor (such as bevacizumab) treatment with immunotherapy(59). Multikinase inhibitors provide only moderate survival benefit(60,61) due to primary resistance and the plasticity of signaling networks(62), and only a subset of patients benefits from addition of immunotherapy in HCC treatment(63)” (page 15).

References

Adhikary, A., S. Chakraborty, M. Mazumdar, S. Ghosh, S. Mukherjee, A. Manna, S. Mohanty, K.K. Nakka, S. Joshi, A. De, S. Chattopadhyay, G. Sa, and T. Das. 2014. Inhibition of epithelial to mesenchymal transition by E-cadherin up-regulation via repression of slug transcription and inhibition of Ecadherin degradation: dual role of scaffold/matrix attachment region-binding protein 1 (SMAR1) in breast cancer cells. *The Journal of biological chemistry*. 289:25431-25444.

Auersperg, N., J. Pan, B.D. Grove, T. Peterson, J. Fisher, S. Maines-Bandiera, A. Somasiri, and C.D. Roskelley. 1999. E-cadherin induces mesenchymal-to-epithelial transition in human ovarian surface epithelium. *Proc Natl Acad Sci U S A*. 96:6249-6254.

Bernal, A., M. McLaughlin, A. Tiwari, F. Cigarroa, and L. Sun. 2024. Abstract 772: Investigation of gender disparity in liver tumor formation using a hydrodynamic tail vein injection mouse model. *Cancer Research*. 84:772-772.

Bigsby, R.M., and A. Caperell-Grant. 2011. The role for estrogen receptor-alpha and prolactin receptor in sex-dependent DEN-induced liver tumorigenesis. *Carcinogenesis*. 32:1162-1166.

Bonakdar, N., A. Schilling, M. Sporrer, P. Lennert, A. Mainka, L. Winter, G. Walko, G. Wiche, B. Fabry, and W.H. Goldmann. 2015. Determining the mechanical properties of plectin in mouse myoblasts and keratinocytes. *Exp Cell Res*. 331:331-337.

Boyault, S., D.S. Rickman, A. de Reynies, C. Balabaud, S. Rebouissou, E. Jeannot, A. Herault, J. Saric, J. Belghiti, D. Franco, P. Bioulac-Sage, P. Laurent-Puig, and J. Zucman-Rossi. 2007. Transcriptome classification of HCC is related to gene alterations and to new therapeutic targets. *Hepatology*. 45:42-52.

Bray, F., M. Laversanne, H. Sung, J. Ferlay, R.L. Siegel, I. Soerjomataram, and A. Jemal. 2024. Global cancer statistics 2022: GLOBOCAN estimates of incidence and mortality worldwide for 36 cancers in 185 countries. *CA Cancer J Clin*. 74:229-263.

Buckup, M., M.A. Rice, E.C. Hsu, F. Garcia-Marques, S. Liu, M. Aslan, A. Bermudez, J. Huang, S.J. Pitteri, and T. Stoyanova. 2021. Plectin is a regulator of prostate cancer growth and metastasis. *Oncogene*. 40:663-676.

Burgstaller, G., M. Gregor, L. Winter, and G. Wiche. 2010. Keeping the vimentin network under control: cell-matrix adhesion-associated plectin 1f affects cell shape and polarity of fibroblasts. *Mol Biol Cell*. 21:3362-3375.

Chintalgattu, V., D.M. Nair, and L.C. Katwa. 2003. Cardiac myofibroblasts: a novel source of vascular endothelial growth factor (VEGF) and its receptors Flt-1 and KDR. *J Mol Cell Cardiol*. 35:277-286. Cuconati, A., C. Mills, C. Goddard, X. Zhang, W. Yu, H. Guo, X. Xu, and T.M. Block. 2013. Suppression of AKT anti-apoptotic signaling by a novel drug candidate results in growth arrest and apoptosis of hepatocellular carcinoma cells. *PLoS One*. 8:e54595.

Du, Y.Q., B. Yuan, Y.X. Ye, F.L. Zhou, H. Liu, J.J. Huang, and Y.F. Wei. 2024. Plumbagin Regulates Snail to Inhibit Hepatocellular Carcinoma Epithelial-Mesenchymal Transition *in vivo* and *in vitro*. *J Hepatocell Carcinoma*. 11:565-580.

Fan, Z.C., J. Yan, G.D. Liu, X.Y. Tan, X.F. Weng, W.Z. Wu, J. Zhou, and X.B. Wei. 2012. Real-time monitoring of rare circulating hepatocellular carcinoma cells in an orthotopic model by *in vivo* flow cytometry assesses resection on metastasis. *Cancer Res*. 72:2683-2691.

Ferrara, N., K. Carver-Moore, H. Chen, M. Dowd, L. Lu, K.S. O'Shea, L. Powell-Braxton, K.J. Hillan, and M.W. Moore. 1996. Heterozygous embryonic lethality induced by targeted inactivation of the VEGF gene. *Nature*. 380:439-442.

Fu, Q., Q. Zhang, Y. Lou, J. Yang, G. Nie, Q. Chen, Y. Chen, J. Zhang, J. Wang, T. Wei, H. Qin, X. Dang, X. Bai, and T. Liang. 2018. Primary tumor-derived exosomes facilitate metastasis by regulating adhesion of circulating tumor cells via SMAD3 in liver cancer. *Oncogene*. 37:6105-6118.

Gerber, H.P., T.H. Vu, A.M. Ryan, J. Kowalski, Z. Werb, and N. Ferrara. 1999. VEGF couples hypertrophic cartilage remodeling, ossification and angiogenesis during endochondral bone formation. *Nat Med*. 5:623-628.

Gnani, D., I. Romito, S. Artuso, M. Chierici, C. De Stefanis, N. Panera, A. Crudele, S. Ceccarelli, E. Carcarino, V. D'Oria, M. Porru, E. Giorda, K. Ferrari, L. Miele, E. Villa, C. Balsano, D. Pasini, C. Furlanello, F. Locatelli, V. Nobili, R. Rota, C. Leonetti, and A. Alisi. 2017. Focal adhesion kinase depletion reduces human hepatocellular carcinoma growth by repressing enhancer of zeste homolog 2. *Cell Death Differ*. 24:889-902.

Gregor, M., S. Osmanagic-Myers, G. Burgstaller, M. Wolfram, I. Fischer, G. Walko, G.P. Resch, A. Jorgl, H. Herrmann, and G. Wiche. 2014. Mechanosensing through focal adhesion-anchored intermediate filaments. *FASEB J*. 28:715-729.

Hiratsuka, S., S. Goel, W.S. Kamoun, Y. Maru, D. Fukumura, D.G. Duda, and R.K. Jain. 2011. Endothelial focal adhesion kinase mediates cancer cell homing to discrete regions of the lungs via E-selectin up-regulation. *Proc Natl Acad Sci U S A*. 108:3725-3730.

Jakab, M., K.H. Lee, A. Uvarovskii, S. Ovchinnikova, S.R. Kulkarni, S. Jakab, T. Rostalski, C. Spegg, S. Anders, and H.G. Augustin. 2024. Lung endothelium exploits susceptible tumor cell states to instruct metastatic latency. *Nat Cancer*. 5:716-730.

Jin, H., C. Wang, G. Jin, H. Ruan, D. Gu, L. Wei, H. Wang, N. Wang, E. Arunachalam, Y. Zhang, X. Deng, C. Yang, Y. Xiong, H. Feng, M. Yao, J. Fang, J. Gu, W. Cong, and W. Qin. 2017. Regulator of Calcineurin 1 Gene Isoform 4, Down-regulated in Hepatocellular Carcinoma, Prevents Proliferation, Migration, and Invasive Activity of Cancer Cells and Metastasis of Orthotopic Tumors by Inhibiting Nuclear Translocation of NFAT1. *Gastroenterology*. 153:799-811 e733.

Jirouskova, M., K. Nepomucka, G. Oyman-Eyrilmez, A. Kalendova, H. Havelkova, L. Sarnova, K. Chalupsky, B. Schuster, O. Benada, P. Miksatkova, M. Kuchar, O. Fabian, R. Sedlacek, G. Wiche, and M. Gregor. 2018. Plectin controls biliary tree architecture and stability in cholestasis. *J Hepatol*. 68:1006-1017.

Katada, K., T. Tomonaga, M. Satoh, K. Matsushita, Y. Tonoike, Y. Kodera, T. Hanazawa, F. Nomura, and Y. Okamoto. 2012. Plectin promotes migration and invasion of cancer cells and is a novel prognostic marker for head and neck squamous cell carcinoma. *J Proteomics*. 75:1803-1815.

Koster, J., S. van Wilpe, I. Kuikman, S.H. Litjens, and A. Sonnenberg. 2004. Role of binding of plectin to the integrin beta4 subunit in the assembly of hemidesmosomes. *Mol Biol Cell*. 15:1211-1223.

Liu, H., Q. Chen, D. Lu, X. Pang, S. Yin, K. Wang, R. Wang, S. Yang, Y. Zhang, Y. Qiu, T. Wang, and H. Yu. 2020. HTBPI, an active phenanthroindolizidine alkaloid, inhibits liver tumorigenesis by targeting Akt. *FASEB J*. 34:12255-12268.

Lu, H.H., S.Y. Lin, R.R. Weng, Y.H. Juan, Y.W. Chen, H.H. Hou, Z.C. Hung, G.A. Oswita, Y.J. Huang, S.Y. Guu, K.H. Khoo, J.Y. Shih, C.J. Yu, and H.C. Tsai. 2020. Fucosyltransferase 4 shapes oncogenic glycoproteome to drive metastasis of lung adenocarcinoma. *EBioMedicine*. 57:102846.

Mathews, S.T., E.P. Plaisance, and T. Kim. 2009. Imaging systems for westerns: chemiluminescence vs. infrared detection. *Methods in molecular biology (Clifton, N.J.)*. 536:499-513.

Osmanagic-Myers, S., M. Gregor, G. Walko, G. Burgstaller, S. Reipert, and G. Wiche. 2006. Plectincontrolled keratin cytoarchitecture affects MAP kinases involved in cellular stress response and migration. *J Cell Biol*. 174:557-568.

Osmanagic-Myers, S., S. Rus, M. Wolfram, D. Brunner, W.H. Goldmann, N. Bonakdar, I. Fischer, S. Reipert, A. Zuzuarregui, G. Walko, and G. Wiche. 2015. Plectin reinforces vascular integrity by mediating crosstalk between the vimentin and the actin networks. *J Cell Sci*. 128:4138-4150.

Pillai-Kastoori, L., A.R. Schutz-Geschwender, and J.A. Harford. 2020. A systematic approach to quantitative Western blot analysis. *Analytical biochemistry*. 593:113608.

Pitot, H.C. 2001. Pathways of progression in hepatocarcinogenesis. *Lancet (London, England)*. 358:859860.

Prechova, M., Z. Adamova, A.L. Schweizer, M. Maninova, A. Bauer, D. Kah, S.M. Meier-Menches, G. Wiche, B. Fabry, and M. Gregor. 2022. Plectin-mediated cytoskeletal crosstalk controls cell tension and cohesion in epithelial sheets. *J Cell Biol*. 221.

Prechova, M., K. Korelova, and M. Gregor. 2023. Plectin. *Curr Biol*. 33:R128-R130.

Qi, L., T. Knifley, M. Chen, and K.L. O'Connor. 2022. Integrin alpha6beta4 requires plectin and vimentin for adhesion complex distribution and invasive growth. *J Cell Sci*. 135.

Romito, I., M. Porru, M.R. Braghini, L. Pompili, N. Panera, A. Crudele, D. Gnani, C. De Stefanis, M. Scarsella, S. Pomella, S. Levi Mortera, E. de Billy, A.L. Conti, V. Marzano, L. Putignani, M. Vinciguerra, C. Balsano, A. Pastore, R. Rota, M. Tartaglia, C. Leonetti, and A. Alisi. 2021. Focal adhesion kinase inhibitor TAE226 combined with Sorafenib slows down hepatocellular carcinoma by multiple epigenetic effects. *J Exp Clin Cancer Res*. 40:364.

Si, T., L. Huang, T. Liang, P. Huang, H. Zhang, M. Zhang, and X. Zhou. 2023. Ruangan Lidan decoction inhibits the growth and metastasis of liver cancer by downregulating miR-9-5p and upregulating PDK4. *Cancer Biol Ther*. 24:2246198.

Strumberg, D., J.W. Clark, A. Awada, M.J. Moore, H. Richly, A. Hendlisz, H.W. Hirte, J.P. Eder, H.J. Lenz, and B. Schwartz. 2007. Safety, pharmacokinetics, and preliminary antitumor activity of sorafenib: a review of four phase I trials in patients with advanced refractory solid tumors. *Oncologist*. 12:426-437.

Tao, Q.F., S.X. Yuan, F. Yang, S. Yang, Y. Yang, J.H. Yuan, Z.G. Wang, Q.G. Xu, K.Y. Lin, J. Cai, J. Yu, W.L. Huang, X.L. Teng, C.C. Zhou, F. Wang, S.H. Sun, and W.P. Zhou. 2015. Aldolase B inhibits metastasis through Ten-Eleven Translocation 1 and serves as a prognostic biomarker in hepatocellular carcinoma. *Mol Cancer*. 14:170.

Vahidnezhad, H., L. Youssefian, N. Harvey, A.R. Tavasoli, A.H. Saeidian, S. Sotoudeh, A. Varghaei, H. Mahmoudi, P. Mansouri, N. Mozafari, O. Zargari, S. Zeinali, and J. Uitto. 2022. Mutation update: The spectra of PLEC sequence variants and related plectinopathies. *Human mutation*. 43:17061731.

Voisin, L., M. Lapouge, M.K. Saba-El-Leil, M. Gombos, J. Javary, V.Q. Trinh, and S. Meloche. 2024. Syngeneic mouse model of YES-driven metastatic and proliferative hepatocellular carcinoma. *Dis Model Mech*. 17.

Wang, D.D., Y. Chen, Z.B. Chen, F.J. Yan, X.Y. Dai, M.D. Ying, J. Cao, J. Ma, P.H. Luo, Y.X. Han, Y. Peng, Y.H. Sun, H. Zhang, Q.J. He, B. Yang, and H. Zhu. 2016. CT-707, a Novel FAK Inhibitor, Synergizes with Cabozantinib to Suppress Hepatocellular Carcinoma by Blocking Cabozantinib-Induced FAK Activation. *Mol Cancer Ther*. 15:2916-2925.

Wang, W., A. Zuidema, L. Te Molder, L. Nahidiazar, L. Hoekman, T. Schmidt, S. Coppola, and A. Sonnenberg. 2020. Hemidesmosomes modulate force generation via focal adhesions. *J Cell Biol*. 219.

Wendt, M.K., M.A. Taylor, B.J. Schiemann, and W.P. Schiemann. 2011. Down-regulation of epithelial cadherin is required to initiate metastatic outgrowth of breast cancer. *Mol Biol Cell*. 22:24232435.

Wenta, T., A. Schmidt, Q. Zhang, R. Devarajan, P. Singh, X. Yang, A. Ahtikoski, M. Vaarala, G.H. Wei, and A. Manninen. 2022. Disassembly of alpha6beta4-mediated hemidesmosomal adhesions promotes tumorigenesis in PTEN-negative prostate cancer by targeting plectin to focal adhesions. *Oncogene*. 41:3804-3820.

Wilhelm, S., C. Carter, M. Lynch, T. Lowinger, J. Dumas, R.A. Smith, B. Schwartz, R. Simantov, and S. Kelley. 2006. Discovery and development of sorafenib: a multikinase inhibitor for treating cancer. *Nat Rev Drug Discov*. 5:835-844.

Wilhelm, S.M., C. Carter, L. Tang, D. Wilkie, A. McNabola, H. Rong, C. Chen, X. Zhang, P. Vincent, M. McHugh, Y. Cao, J. Shujath, S. Gawlak, D. Eveleigh, B. Rowley, L. Liu, L. Adnane, M. Lynch, D. Auclair, I. Taylor, R. Gedrich, A. Voznesensky, B. Riedl, L.E. Post, G. Bollag, and P.A. Trail. 2004. BAY 43-9006 exhibits broad spectrum oral antitumor activity and targets the RAF/MEK/ERK pathway and receptor tyrosine kinases involved in tumor progression and angiogenesis. *Cancer Res*. 64:7099-7109.

Xu, R., S. He, D. Ma, R. Liang, Q. Luo, and G. Song. 2022. Plectin Downregulation Inhibits Migration and Suppresses Epithelial Mesenchymal Transformation of Hepatocellular Carcinoma Cells via ERK1/2 Signaling. *Int J Mol Sci*. 24.

You, A., M. Cao, Z. Guo, B. Zuo, J. Gao, H. Zhou, H. Li, Y. Cui, F. Fang, W. Zhang, T. Song, Q. Li, X. Zhu, H. Yin, H. Sun, and T. Zhang. 2016. Metformin sensitizes sorafenib to inhibit postoperative recurrence and metastasis of hepatocellular carcinoma in orthotopic mouse models. *J Hematol Oncol*. 9:20.

Yuzugullu, H., K. Benhaj, N. Ozturk, S. Senturk, E. Celik, A. Toylu, N. Tasdemir, M. Yilmaz, E. Erdal, K.C. Akcali, N. Atabey, and M. Ozturk. 2009a. Canonical Wnt signaling is antagonized by noncanonical Wnt5a in hepatocellular carcinoma cells. *Molecular Cancer*. 8:90.

Yuzugullu, H., K. Benhaj, N. Ozturk, S. Senturk, E. Celik, A. Toylu, N. Tasdemir, M. Yilmaz, E. Erdal, K.C. Akcali, N. Atabey, and M. Ozturk. 2009b. Canonical Wnt signaling is antagonized by noncanonical Wnt5a in hepatocellular carcinoma cells. *Mol Cancer*. 8:90.

Zhao, J., Y. Hou, C. Yin, J. Hu, T. Gao, X. Huang, X. Zhang, J. Xing, J. An, S. Wan, and J. Li. 2020. Upregulation of histamine receptor H1 promotes tumor progression and contributes to poor prognosis in hepatocellular carcinoma. *Oncogene*. 39:1724-1738.

Zheng, H., Y. Yang, C. Ye, P.P. Li, Z.G. Wang, H. Xing, H. Ren, and W.P. Zhou. 2018. Lamp2 inhibits epithelial-mesenchymal transition by suppressing Snail expression in HCC. *Oncotarget*. 9:3024030252.